# Blind Inverse Game Theory: Jointly Decoding Rewards and Rationality in Entropy-Regularized Competitive Games

## Abstract

Inverse Game Theory (IGT) methods based on the entropy-regularized Quantal Response Equilibrium (QRE) offer a tractable approach for competitive settings, but critically assume the agents' rationality parameter (temperature $\tau$) is known a priori. When $\tau$ is unknown, a fundamental scale ambiguity emerges that couples $\tau$ with the reward parameters ($\theta$), making them statistically unidentifiable. We introduce Blind-IGT, the first statistical framework to jointly recover both $\theta$ and $\tau$ from observed behavior. We analyze this bilinear inverse problem and establish necessary and sufficient conditions for unique identification by introducing a normalization constraint that resolves the scale ambiguity. We propose an efficient Normalized Least Squares (NLS) estimator and prove it achieves the optimal $\mathcal{O}(N^{-1/2})$ convergence rate for joint parameter recovery. When strong identifiability conditions fail, we provide partial identification guarantees through confidence set construction. We extend our framework to Markov games and demonstrate optimal convergence rates with strong empirical performance even when transition dynamics are unknown.

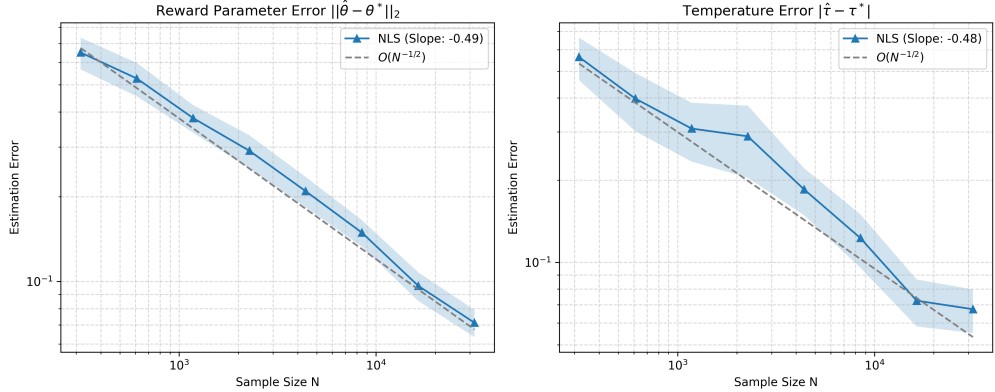

Figure 1: Convergence analysis of the NLS algorithm in Matrix Games. Log-log plots demonstrate that the estimation errors for both the reward parameters ($\theta$, Left) and the unknown temperature ($\tau$, Right) decrease at the optimal parametric rate of $\mathcal{O}(N^{-1/2})$ (dashed line), confirming the statistical efficiency of the Blind-IGT framework.

## 1 INTRODUCTION

The statistical inference of underlying objectives from observed strategic behavior is a fundamental challenge spanning artificial intelligence, economics, and multi-agent systems (Ng & Russell, 2000; Arora & Doshi, 2021). In multi-agent environments, this problem is formalized as Inverse Game Theory (IGT) (Kuleshov & Schrijvers, 2015; Yu et al., 2019), where the goal is to identify the payoff structures that rationalize observed equilibrium play.

Recovering rewards in competitive settings, particularly two-player zero-sum games, presents significant statistical difficulties. Standard IGT approaches based on the Nash Equilibrium (NE) (Nash, 1951) concept are generally ill-posed, leading to fundamental identification issues (Ahuja & Orlin, 2001; Metelli et al., 2021), equilibrium multiplicity, and computational intractability (Wang & Klabjan, 2018; Wu et al., 2022).

To address this primary identification challenge, the literature has increasingly adopted models of bounded rationality, notably through entropy regularization (Mertikopoulos & Sandholm, 2016). This smooths the agents' best responses, resulting in a unique equilibrium known as the Quantal Response Equilibrium (QRE) (McKelvey & Palfrey, 1995). The QRE framework has facilitated improved identifiability in single-agent Inverse Reinforcement Learning (IRL) (Cao et al., 2021; Rolland et al., 2022) and enabled efficient reward recovery in zero-sum IGT by transforming the problem into a tractable system of linear equations (Liao et al., 2025).

## 1.1 The Challenge: Unknown Rationality and Scale Ambiguity

Despite these advances, current QRE-based IGT methods rely on the critical and often unrealistic assumption that the entropy regularization parameter, denoted as the temperature $\tau > 0$, is known exactly *a priori*. This parameter quantifies the agents' level of stochasticity or bounded rationality.

The assumption of a known $\tau$ is problematic for statistical inference. In practice, the agents' rationality is unknown and must be estimated from the data. Critically, if $\tau$ is unknown, a fundamental identification problem re-emerges. The reward parameters ($\theta$) and the temperature ($\tau$) are inextricably coupled in the QRE definition, which depends only on the ratio $Q/\tau$. This results in an inherent **multiplicative scale ambiguity**: if a pair $(\theta, \tau)$ explains the observed behavior, then any scaled pair $(k\theta, k\tau)$ for $k > 0$ provides an identical explanation. This ambiguity constitutes a fundamental roadblock because the true parameters become statistically **unidentifiable**. Intuitively, we cannot distinguish between a highly rational agent (small $\tau$) playing a low-stakes game (small $\theta$) and a highly irrational agent (large $\tau$) playing a high-stakes game (large $\theta$). Consequently, an incorrect specification of $\tau$ leads to a systematic bias in the scale of the recovered rewards, leading to model misspecification and invalidating theoretical guarantees.

## 1.2 Our Contributions

To address this fundamental gap, we introduce *Blind Inverse Game Theory* (Blind-IGT), the first statistical framework enabling the simultaneous recovery of reward parameters ($\theta$) and the unknown temperature ($\tau$) from observed equilibrium behavior in entropy-regularized zero-sum games. This requires solving a challenging bilinear inverse problem. By introducing a normalization constraint on the rewards, we resolve the inherent scale ambiguity and provide a comprehensive theoretical analysis. Our main contributions are as follows.

**First**, we provide a comprehensive identifiability analysis where we formally characterize the multiplicative scale ambiguity, establish necessary and sufficient conditions for unique joint identification (Theorem 1), and provide a rigorous framework for partial identification via confidence sets when these conditions fail (Proposition 1).

**Second**, we propose an efficient Normalized Least Squares (NLS) estimator with optimal rates (Algorithm 1), providing rigorous finite-sample guarantees that prove NLS achieves the optimal parametric convergence rate of $\mathcal{O}(N^{-1/2})$ for the joint recovery problem (Theorem 2).

**Third**, we develop a robust extension to Markov games (Theorem 3) and empirically demonstrate that NLS tracks the optimal convergence rates even when transition dynamics are unknown and estimated from data (Section 6.4).

**Finally**, extensive simulations confirm the statistical efficiency of NLS, its necessity when $\tau$ is unknown, and its robustness to misspecification.

## 2 RELATED WORK

Our work lies at the intersection of inverse optimization, inverse reinforcement learning, and the statistical analysis of entropy-regularized games.

**Inverse Optimization (IO) and Inverse Reinforcement Learning (IRL).** IO aims to infer objective functions from observed decisions (Ahuja & Orlin, 2001; Chan et al., 2023). IRL specifically focuses on recovering reward functions in sequential decision-making (Ng & Russell, 2000). A prominent approach is Maximum Entropy IRL (MaxEnt-IRL) (Ziebart et al., 2008; Wulfmeier et al., 2016; Finn et al., 2016).

Recent theoretical work has analyzed reward identifiability in entropy-regularized settings. Cao et al. (2021) and Rolland et al. (2022) demonstrated improved identifiability in single-agent scenarios. Crucially, this prior work addresses an *additive* ambiguity inherent in the Bellman equation (potential shaping). In contrast, our work tackles a *multiplicative* scale ambiguity arising from the unknown temperature in a competitive setting. These represent distinct mathematical challenges.

**Inverse Game Theory (IGT).** IGT extends IRL to multi-agent systems, aiming to recover the underlying payoff structures that rationalize observed strategic interactions (Vorobeychik et al., 2007; Arora & Doshi, 2021). The predominant approach in the literature assumes that the observed behavior corresponds to a standard equilibrium concept, typically the Nash Equilibrium (NE) (Lin et al., 2018; Kuleshov & Schrijvers, 2015) or the Correlated Equilibrium (CE) (Waugh et al., 2011). However, these approaches often struggle with non-uniqueness and computational intractability, frequently relying on complex bilevel optimization as discussed in existing work (Wang & Klabjan, 2018; Wu et al., 2022; Konstantakopoulos et al., 2018).

**Entropy Regularization and QRE.** The Quantal Response Equilibrium (QRE) (McKelvey & Palfrey, 1995) models agents playing noisy best responses. Unlike the idealized Nash Equilibrium (NE), QRE provides smooth, unique equilibrium predictions, which facilitates tractable analysis (Mertikopoulos & Sandholm, 2016) and often better fits empirical data from human behavior (Goeree et al., 2016; Camerer, 2003). The temperature parameter $\tau$ explicitly controls the degree of bounded rationality, allowing QRE to interpolate between uniform random play (as $\tau \to \infty$) and the NE (as $\tau \to 0$). This connection makes QRE a central concept in modern multi-agent reinforcement learning (MARL), where entropy regularization is widely employed to stabilize training and encourage exploration (Haarnoja et al., 2018; Cen et al., 2021; Guan et al., 2021; Ahmed et al., 2019; Zhan et al., 2023).

**Relation to prior QRE-based IGT.** The work most closely related to ours is Liao et al. (2025), which established a framework for IGT in entropy-regularized zero-sum games based on the linearized QRE constraints we utilize. Our work directly addresses the fundamental limitation of their analysis (the assumption of a known $\tau$) by developing a "blind" framework. We provide a methodology (NLS) for unique point identification of both rewards and the unknown temperature via normalization. Furthermore, we extend the partial identification analysis of Liao et al. (2025) to the bilinear setting where $\tau$ is unknown.

**Bilinear Inverse Problems.** The structure of Blind-IGT is a bilinear inverse problem, where we seek two vectors whose product explains the observations. Such problems arise in areas like blind deconvolution (Ahmed et al., 2014) and self-calibration (Ling & Strohmer, 2015). They are generally non-convex. Our approach leverages the specific structure of the QRE constraints and the normalization constraint to derive an efficient and provably correct estimation method.

## 3 PRELIMINARIES: ENTROPY-REGULARIZED ZERO-SUM GAMES

We formalize the setting of entropy-regularized two-player zero-sum games and the QRE solution concept.

### 3.1 Matrix Games

A two-player zero-sum matrix game is defined by a triple $(\mathcal{A}, \mathcal{B}, Q)$, where $\mathcal{A} = \{1, \ldots, m\}$ and $\mathcal{B} = \{1, \ldots, n\}$ are the finite action sets. $Q \in \mathbb{R}^{m \times n}$ is the payoff matrix.

**Entropy Regularization.** We study the game under entropy regularization (Mertikopoulos & Sandholm, 2016). The regularized game is formulated as:

$$\max_{\mu \in \Delta(\mathcal{A})} \min_{\nu \in \Delta(\mathcal{B})} \left\{ \mu^\top Q \nu + \tau \mathcal{H}(\mu) - \tau \mathcal{H}(\nu) \right\}, \tag{1}$$

where $\mu \in \Delta(\mathcal{A})$ and $\nu \in \Delta(\mathcal{B})$ are the mixed strategies for each player, $\tau > 0$ is the regularization parameter (temperature), and $\mathcal{H}(\pi) = -\sum_i \pi_i \log(\pi_i)$ is the Shannon entropy (Shannon, 1948).

**Quantal Response Equilibrium (QRE).** The optimization problem equation 1 admits a unique solution $(\mu^*, \nu^*)$, known as the QRE (McKelvey & Palfrey, 1995). The QRE is characterized by the following fixed-point equations (logit responses):

$$\mu^*(a) = \frac{\exp(Q(a, \cdot)\nu^*/\tau)}{\sum_{a' \in \mathcal{A}} \exp(Q(a', \cdot)\nu^*/\tau)}, \tag{2}$$

$$\nu^*(b) = \frac{\exp(-Q(\cdot, b)^\top \mu^*/\tau)}{\sum_{b' \in \mathcal{B}} \exp(-Q(\cdot, b')^\top \mu^*/\tau)}. \tag{3}$$

These non-linear equations can be linearized by taking the logarithm and normalizing with respect to a reference action (e.g., action 1). This transformation (detailed in Appendix B.1) yields the following system of $m + n - 2$ linear constraints:

$$\begin{cases} (Q(a, \cdot) - Q(1, \cdot))\nu^* = \tau \cdot \log \frac{\mu^*(a)}{\mu^*(1)}, & \forall a \in \mathcal{A} \setminus \{1\} \\ (Q(\cdot, 1) - Q(\cdot, b))^\top \mu^* = \tau \cdot \log \frac{\nu^*(b)}{\nu^*(1)}, & \forall b \in \mathcal{B} \setminus \{1\} \end{cases}. \tag{4}$$

### 3.2 Markov Games

The framework extends to dynamic settings modeled as two-player zero-sum Markov games (MGs). An infinite-horizon discounted MG is defined by the tuple $(\mathcal{S}, \mathcal{A}, \mathcal{B}, r, P, \gamma)$. $\mathcal{S}$ is the state space. $\mathcal{A}$ and $\mathcal{B}$ are the action spaces for player 1 and 2, respectively. $r : \mathcal{S} \times \mathcal{A} \times \mathcal{B} \to \mathbb{R}$ is the reward function (payoff for player 1), $P(\cdot|s, a, b)$ is the transition probability distribution over the next state $s'$ given the current state $s$ and joint action $(a, b)$. $\gamma \in [0, 1)$ is the discount factor.

Agents interact by choosing stationary policies $\mu : \mathcal{S} \to \Delta(\mathcal{A})$ and $\nu : \mathcal{S} \to \Delta(\mathcal{B})$. In the entropy-regularized setting, the V-function $V^{\mu,\nu}(s)$ and the Q-function $Q^{\mu,\nu}(s, a, b)$ are defined recursively via the Bellman equations:

$$Q^{\mu,\nu}(s, a, b) = r(s, a, b) + \gamma \mathbb{E}_{s' \sim P(\cdot|s,a,b)}[V^{\mu,\nu}(s')], \tag{5}$$

$$\begin{aligned} V^{\mu,\nu}(s) &= \mu(s)^\top Q^{\mu,\nu}(s)\nu(s) \\ &\quad + \tau \mathcal{H}(\mu(s)) - \tau \mathcal{H}(\nu(s)). \end{aligned} \tag{6}$$

The QRE $(\mu^*, \nu^*)$ is the unique solution to the minimax problem $V^*(s) = \max_\mu \min_\nu V^{\mu,\nu}(s)$. Similar to matrix games, the QRE policies at each state $s$ satisfy the fixed-point equations defined by equation 2 and equation 3, using the Q-function $Q^*(s)$ in place of the payoff matrix $Q$.

## 4 BLIND INVERSE GAME THEORY IN MATRIX GAMES

In this section, we introduce the Blind-IGT problem for matrix games, analyze the inherent ambiguity, establish identifiability conditions, propose the Normalized Least Squares estimator, and analyze the case of partial identification.

### 4.1 Problem Formulation and Ambiguity

The objective of Blind-IGT is to recover the payoff matrix $Q$ and the temperature $\tau > 0$ from observations of the equilibrium behavior $(\mu^*, \nu^*)$. We leverage the standard assumption of linear parameterization.

**Assumption 1** (Linear Payoff Functions)**.** There exists a known feature map $\phi : \mathcal{A} \times \mathcal{B} \to \mathbb{R}^d$ and an unknown parameter vector $\theta^* \in \mathbb{R}^d$ such that $Q(a, b) = \langle \phi(a, b), \theta^* \rangle$. We assume bounded features, $\|\phi(a, b)\|_2 \le L$.

Under Assumption 1, the linearized QRE constraints in equation 4 can be rewritten in a compact matrix-vector form. We define the feature matrices $A(\nu) \in \mathbb{R}^{(m-1) \times d}$ and $B(\mu) \in \mathbb{R}^{(n-1) \times d}$, and the log-ratio vectors $c(\mu) \in \mathbb{R}^{m-1}$ and $d(\nu) \in \mathbb{R}^{n-1}$.

**System Construction Details.** The feature matrices are defined row-wise. For $a \in \{2, \ldots, m\}$ (row $a-1$) and $b \in \{2, \ldots, n\}$ (row $b-1$):

$$A(\nu)_{a-1} = \sum_{b' \in \mathcal{B}} \nu(b')(\phi(a, b') - \phi(1, b'))^\top,$$

$$B(\mu)_{b-1} = \sum_{a' \in \mathcal{A}} \mu(a')(\phi(a', 1) - \phi(a', b))^\top.$$

The log-ratio vectors are defined element-wise:

$$c(\mu)_{a-1} = \log(\mu(a)/\mu(1)), \quad d(\nu)_{b-1} = \log(\nu(b)/\nu(1)).$$

Let $X(\mu, \nu) = [A(\nu)^\top, B(\mu)^\top]^\top$ and $y(\mu, \nu) = [c(\mu)^\top, d(\nu)^\top]^\top$. The QRE constraints form a bilinear system of equations:

$$X(\mu^*, \nu^*)\theta^* = \tau^* \cdot y(\mu^*, \nu^*). \tag{7}$$

**The Scale Ambiguity.** The system equation 7 highlights the fundamental identification challenge. If $(\theta^*, \tau^*)$ is a solution, then any scaled pair $(k\theta^*, k\tau^*)$ for $k > 0$ is also a solution that induces the exact same QRE $(\mu^*, \nu^*)$.

## 4.2 Identifiability Conditions

To achieve strong identification, we must introduce a constraint that breaks the scale homogeneity. We assume that the scale of the underlying reward parameters $\theta^*$ is known.

**Assumption 2** (Normalization Constraint)**.** The true reward parameter $\theta^*$ satisfies $\|\theta^*\|_2 = C$ for some known constant $C > 0$. Without loss of generality, we assume $C = 1$.

Assumption 2 is necessary for the exact recovery of the scales of both $\theta^*$ and $\tau^*$.

*Remark* 1 (Robustness to Misspecification). Crucially, we demonstrate in Section 4.5 that our approach is highly robust even if the constant $C$ is misspecified. The NLS algorithm still accurately recovers the *direction* of $\theta^*$ (the relative importance of features), which is often sufficient for behavioral modeling.

We now establish the necessary and sufficient conditions for the unique joint identification of the pair $(\theta^*, \tau^*)$.

**Theorem 1** (Identifiability of Blind-IGT)**.** *Under Assumptions 1 and 2 (with $C = 1$), the pair $(\theta^*, \tau^*)$ is uniquely identifiable from the QRE $(\mu^*, \nu^*)$ if and only if the following two conditions hold:*

1. ***Rank Condition:*** *The matrix $X(\mu^*, \nu^*)$ has full column rank, i.e., $rank(X(\mu^*, \nu^*)) = d$.*

2. ***Non-Uniformity Condition:*** *The QRE is not the uniform distribution, i.e., $y(\mu^*, \nu^*) \ne 0$.*

*Proof Sketch (Appendix C.1).* Sufficiency relies on the Rank Condition ensuring a unique direction for $\theta^*$ via the Moore-Penrose inverse (Penrose, 1955). The Normalization Constraint then uniquely determines the scale (and thus $\tau^*$), provided the Non-Uniformity Condition holds (ensuring $\theta^* \ne 0$). Necessity is shown by constructing counterexamples when either condition fails. □

### 4.3 Estimation via Normalized Least Squares

We propose an algorithm to estimate $(\theta^*, \tau^*)$ from $N$ i.i.d. samples $\{(a^k, b^k)\}_{k=1}^N$ drawn from the QRE $(\mu^*, \nu^*)$. We first estimate the QRE using the empirical frequency estimators $(\hat{\mu}, \hat{\nu})$, and construct the estimated matrices $\hat{X} = X(\hat{\mu}, \hat{\nu})$ and $\hat{y} = y(\hat{\mu}, \hat{\nu})$.

We propose the Normalized Least Squares (NLS) estimator. The complete pseudocode is provided in Algorithm 1 in Appendix F. The core idea is to first estimate the ratio $\theta^*/\tau^*$ using standard least squares, and then use the normalization constraint to disentangle the two parameters.

### 4.4 Theoretical Guarantees

We analyze the finite-sample performance of the NLS estimator. We establish that $(\hat{\theta}, \hat{\tau})$ converge to the true parameters $(\theta^*, \tau^*)$ at the optimal parametric rate of $\mathcal{O}(N^{-1/2})$.

To ensure the stability of the log-ratio vectors $y(\mu, \nu)$, we require that the QRE probabilities are uniformly bounded away from zero.

**Assumption 3** (Soft-Min Gap)**.** There exists a constant $\xi > 0$ such that the QRE probabilities satisfy $\min_{a,b}\{\mu^*(a), \nu^*(b)\} \geq \xi$.

Let $\sigma_{\min}(X^*)$ denote the minimum singular value of the true matrix $X^*$.

**Theorem 2** (Finite Sample Bounds for Blind-IGT)**.** *Let the identifiability conditions of Theorem 1 hold. Under Assumptions 1, 2 (with $C = 1$), and 3. Given $N$ samples from the QRE $(\mu^*, \nu^*)$, let $(\hat{\theta}, \hat{\tau})$ be the output of Algorithm 1. For any $\delta \in (0, 1)$, if $N$ is sufficiently large (see Appendix C.3 for precise condition), then with probability at least $1 - \delta$:*

$$\|\hat{\theta} - \theta^*\|_2 \leq C_\theta \cdot \sqrt{\frac{(m+n)\log(1/\delta)}{N}}, \tag{8}$$

$$|\hat{\tau} - \tau^*| \leq C_\tau \cdot \sqrt{\frac{(m+n)\log(1/\delta)}{N}}. \tag{9}$$

*The constants $C_\theta$ and $C_\tau$ depend on the problem parameters $(L, \xi, \tau^*, \sigma_{\min}(X^*))$.*

*Proof Sketch (Appendix C.3).* The proof utilizes concentration inequalities (Lemma 1) to bound the perturbation of the empirical system $(\hat{X}, \hat{y})$ (Lemma 2). We then apply standard perturbation theory for least squares (Higham, 2002) to bound the error in the intermediate solution $\hat{\theta}_{LS}$. Finally, a stability analysis of the normalization step (Lemma 3) confirms the optimal rate for both $\hat{\theta}$ and $\hat{\tau}$. □

### 4.5 Robustness to Misspecified Normalization

We analyze the behavior of the NLS estimator when the normalization constant $C$ is misspecified, validating Remark 1.

Suppose the true normalization is $C_{\text{true}}$ but the algorithm is run with $C_{\text{assumed}}$. The NLS algorithm first estimates $\hat{\theta}_{LS} \approx \theta^*/\tau^* = (C_{\text{true}}/\tau^*)\theta^*_{\text{dir}}$, where $\theta^*_{\text{dir}}$ is the normalized direction.

The algorithm then estimates the temperature as:

$$\hat{\tau} = \frac{C_{\text{assumed}}}{\|\hat{\theta}_{LS}\|_2} \approx \frac{C_{\text{assumed}}}{C_{\text{true}}/\tau^*} = \tau^* \cdot \left(\frac{C_{\text{assumed}}}{C_{\text{true}}}\right). \tag{10}$$

The estimated temperature scales linearly with the misspecification ratio. The final reward estimate is $\hat{\theta} = \hat{\tau}\hat{\theta}_{LS}$. Crucially, the direction of $\hat{\theta}$ is identical to the direction of $\hat{\theta}_{LS}$, which is an unbiased estimator of the true direction $\theta^*_{\text{dir}}$.

We validate this empirically (Figure 2) using a matrix game setup ($m = n = 20, d = 8$). We set $C_{\text{true}} = 5.0$ and $\tau^* = 2.0$, and vary the misspecification ratio from 0.1 to 10. The directional error remains consistently low and stable, demonstrating that the relative importance of the reward features can be reliably recovered even without knowledge of the true utility scale.

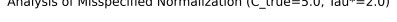

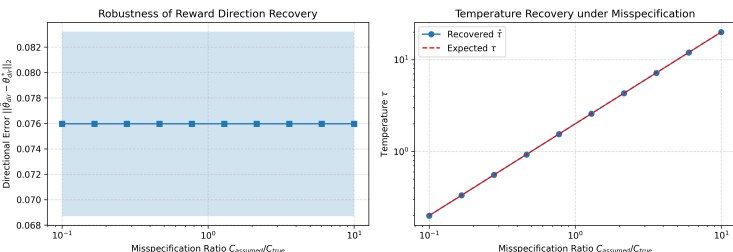

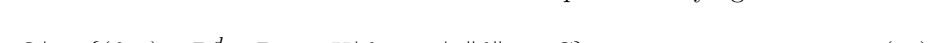

Figure 2: Robustness to Misspecified Normalization Constant $C$. (Left) The directional error of the reward parameters remains low and stable. (Right) The estimated temperature scales linearly with the misspecification ratio, matching the theoretical expectation (dashed red line).

## 4.6 Partial Identification and Confidence Sets

When the Rank Condition of Theorem 1 fails (i.e., $\text{rank}(X^*) < d$), the parameters $(\theta^*, \tau^*)$ are not uniquely identifiable, even with the normalization constraint. In this scenario, we shift our focus from point estimation to *partial identification*, aiming to characterize the set of all parameters consistent with the observed QRE.

**The Identified Set.** We define the identified set $\Theta^*$ as the set of all pairs satisfying the Blind-IGT constraints:

$$\Theta^* = \{(\theta, \tau) \in \mathbb{R}^d \times \mathbb{R}_{>0} : X^*\theta = \tau y^*, \|\theta\|_2 = C\}. \tag{11}$$

When the Rank Condition fails, this set contains multiple elements. Unlike the known-$\tau$ case where the feasible set is an affine subspace (Liao et al., 2025), here $\Theta^*$ is the intersection of a linear subspace (defined by the bilinear constraint) and a non-convex constraint (the normalization sphere).

**Confidence Set Construction.** In a finite-sample setting, we aim to construct a confidence set $\hat{\Theta}_N$ that contains the true identified set $\Theta^*$ with high probability. We define this set based on the residual of the empirical bilinear system, introducing a threshold parameter $\kappa_N > 0$:

$$\hat{\Theta}_N = \{(\theta, \tau) : \|\hat{X}\theta - \tau\hat{y}\|_2^2 \leq \kappa_N, \|\theta\|_2 = C, \tau > 0\}. \tag{12}$$

To establish the validity of this confidence set, we require an assumption that the true temperature is bounded. This is necessary because the impact of the noise in $\hat{y}$ on the residual scales with $\tau$.

**Assumption 4** (Bounded Temperature). *The true temperature parameter is bounded, i.e., $\tau^* \leq \tau_{max}$ for some known constant $\tau_{max} > 0$.*

We can now establish the coverage guarantee for the confidence set $\hat{\Theta}_N$ by appropriately choosing the threshold $\kappa_N$.

**Proposition 1** (Coverage of the Confidence Set). *Under Assumptions 1-3 and Assumption 4. Let $\epsilon_N = \mathcal{O}(\sqrt{(m+n)\log(1/\delta)/N})$ be the policy estimation error rate (Lemma 1). If we choose the threshold $\kappa_N$ such that*

$$\kappa_N \geq (C_X C + C_Y \tau_{max})^2 \cdot \epsilon_N^2,$$

*where $C_X, C_Y$ are the perturbation constants (Lemma 2), then with probability at least $1 - \delta$:*

$$\Theta^* \subseteq \hat{\Theta}_N.$$

*Proof Sketch (Appendix C.4).* We analyze the empirical residual $\|\hat{X}\theta^* - \tau^*\hat{y}\|_2$ for any $(\theta^*, \tau^*) \in \Theta^*$. Since $X^*\theta^* = \tau^* y^*$, the residual depends only on the estimation errors $(\hat{X} - X^*)$ and $(\hat{y} - y^*)$. By applying the triangle inequality and leveraging the perturbation bounds (Lemma 2) along with Assumption 4, we establish an upper bound on the residual, which dictates the choice of $\kappa_N$. □

Proposition 1 guarantees that the constructed confidence set $\hat{\Theta}_N$ reliably captures all feasible reward and temperature parameters consistent with the observed behavior, providing a comprehensive solution to the Blind-IGT problem even when strong identifiability fails.

## 5 EXTENSION TO MARKOV GAMES

We extend the Blind-IGT framework to entropy-regularized zero-sum Markov games.

### 5.1 Problem Formulation and Identifiability

In the Markov game setting, we aim to recover the reward function $r$ and the temperature $\tau^*$ from observed trajectories. We adopt an assumption common in the theoretical analysis of RL with function approximation (Jin et al., 2020).

**Assumption 5** (Linear Q-Functions). There exists a feature map $\phi : \mathcal{S} \times \mathcal{A} \times \mathcal{B} \to \mathbb{R}^d$ such that the unique QRE Q-function $Q^*(s, a, b)$ can be represented as $Q^*(s, a, b) = \langle \phi(s, a, b), \theta^* \rangle$. We assume a normalization constraint: $\|\theta^*\|_2 = R$ for a known constant $R$. We assume bounded features $\|\phi\|_2 \leq L$.

We adopt Assumption 5 as a necessary starting point for rigorous theoretical analysis in dynamic settings. While our primary theoretical analysis (Theorem 3) assumes the transition dynamics $P$ are known for analytical tractability, Algorithm 2 is readily applicable when $P$ is estimated from data, as demonstrated empirically in Section 6.4.

At each state $s$, the QRE policies $(\mu^*(s), \nu^*(s))$ satisfy the linearized constraints analogous to equation 4. We define the local matrices $X(s)$ and vectors $y(s)$. The constraints are:

$$X(s; \mu^*, \nu^*)\theta^* = \tau^* \cdot y(s; \mu^*, \nu^*), \quad \forall s \in \mathcal{S}. \tag{13}$$

**Identifiability.** To identify $\theta^*$, we aggregate the constraints across all states. Let $X^*$ be the matrix formed by stacking $X(s)^*$ vertically, and similarly for $y^*$.

**Proposition 2** (Identifiability in Markov Games). *Under Assumption 5 (with $R = 1$), the pair $(\theta^*, \tau^*)$ is uniquely identifiable if the aggregated matrix $X^*$ has full column rank (rank $d$) and $y^* \neq 0$.*

### 5.2 Estimation Algorithm and Guarantees

We adapt the NLS algorithm (Algorithm 2). We first estimate the policies $(\hat{\mu}, \hat{\nu})$ from the trajectories. If the dynamics are unknown, we estimate $\hat{P}$ from the observed transitions. We construct the aggregated matrices $\hat{X}$ and $\hat{y}$, and apply the NLS procedure. The full algorithm is detailed in Algorithm 2 in Appendix F. Once $\hat{\theta}$ and $\hat{\tau}$ are estimated, we recover the rewards using the Bellman equation equation 5.

We provide theoretical guarantees for Algorithm 2 assuming known dynamics, requiring sufficient coverage of the state space.

**Assumption 6** (Coverage). Let $d^*(s)$ be the stationary distribution induced by the QRE. Assume $d^*(s) \geq C_{\min} > 0$ for all $s \in \mathcal{S}$.

**Theorem 3** (Sample Complexity for MGs). *Under Assumptions 3 (extended to MGs), 5 (with $R = 1$), and 6, and assuming the identifiability conditions of Proposition 2 hold and $P$ is known. Let $K$ be the total number of state-action samples. If $K$ is sufficiently large, then with probability at least $1 - \delta$:*

$$\|\hat{\theta} - \theta^*\|_2 = \mathcal{O}\left(\sqrt{\frac{|\mathcal{S}|(m+n)\log(|\mathcal{S}|/\delta)}{K}}\right), \tag{14}$$

$$|\hat{\tau} - \tau^*| = \mathcal{O}\left(\sqrt{\frac{|\mathcal{S}|(m+n)\log(|\mathcal{S}|/\delta)}{K}}\right), \tag{15}$$

$$\|\hat{r} - r^*\|_\infty = \mathcal{O}\left(\sqrt{\frac{|\mathcal{S}|(m+n)\log(|\mathcal{S}|/\delta)}{K}}\right). \tag{16}$$

*The constants depend on $L, \xi, \tau^*, \sigma_{\min}(X^*), C_{\min}, \gamma$.*

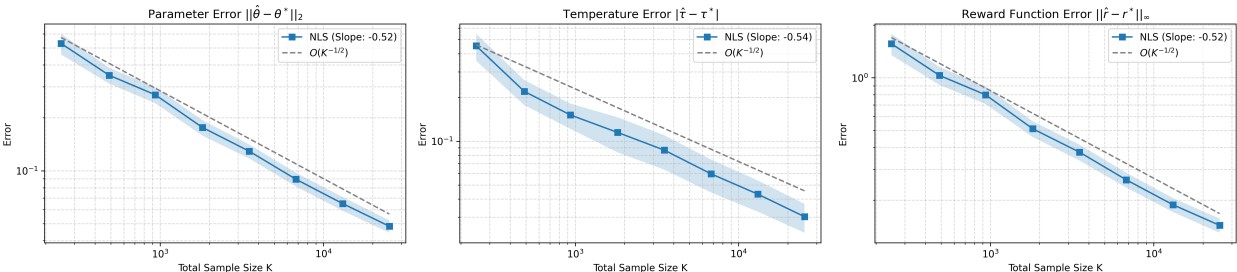

Figure 3: Convergence analysis in Markov Games (Theorem 3). The empirical slopes for Q-parameters ($\theta$), temperature ($\tau$), and the recovered reward function ($r$) closely match the theoretical $\mathcal{O}(K^{-1/2})$ rate.

*Proof Sketch (Appendix D.2).* We first establish uniform convergence of policy estimates across states. The bounds for $\hat{\theta}$ and $\hat{\tau}$ follow the analysis of Theorem 2 applied to the aggregated system. The key challenge is bounding the reward recovery error. This is achieved by establishing the Lipschitz continuity of the V-function with respect to its components $(Q, \mu, \nu, \tau)$, thereby controlling the error propagation through the Bellman equation. □

*Remark* 2 (Unknown Dynamics)*.* Theorem 3 provides guarantees assuming access to the true dynamics $P$. When $P$ is unknown and estimated as $\hat{P}$ (Algorithm 2), the error analysis must account for the uncertainty in $\hat{P}$. However, we demonstrate empirically in Section 6.4 that the optimal $\mathcal{O}(K^{-1/2})$ rate is robustly achieved when estimating $P$ from the same trajectories.

## 6 EXPERIMENTS

We conduct a comprehensive empirical evaluation of the Blind-IGT framework and the NLS algorithm. Our experiments aim to: (1) validate the theoretical convergence rates; (2) demonstrate the necessity of Blind-IGT when the temperature is unknown; and (3) assess robustness to model misspecification and unknown dynamics. (See Appendix E for detailed setup).

### 6.1 Experimental Setup

**Matrix Games.** We generate random zero-sum matrix games with $m = n = 10$ actions and feature dimension $d = 5$. Features $\phi(a, b)$ and the true parameter $\theta^*$ are sampled from a standard normal distribution, with $\theta^*$ normalized to $\|\theta^*\|_2 = 1$. The true temperature is set to $\tau^* = 2.0$.

**Markov Games.** We simulate a zero-sum Markov Game with $|\mathcal{S}| = 8$ states, $m = n = 5$ actions, and discount factor $\gamma = 0.9$. We adopt the Linear Q-function parameterization (Assumption 5) with $d = 6$. The true temperature is set to $\tau^* = 1.5$.

All results are averaged over 50 independent trials.

### 6.2 Validation of Convergence Rates

We validate the finite-sample guarantees established in Theorem 2 and Theorem 3.

**Matrix Games Results.** Figure 1 shows the estimation errors for $\hat{\theta}$ and $\hat{\tau}$. On a log-log scale, the empirical slopes are -0.49 for $\theta$ and -0.48 for $\tau$. These results closely match the theoretical prediction of -0.5.

**Markov Games Results.** Figure 3 presents the results for the dynamic setting (assuming known P). The empirical slopes for the Q-parameters ($\hat{\theta}$) and the recovered reward function ($\hat{r}$) are -0.52, and -0.54 for temperature ($\hat{\tau}$). This strongly validates Theorem 3.

Table 1: Comparison of Blind-IGT (NLS) vs. Standard IGT Liao et al. (2025) with known/misspecified temperature ($\tau^* = 2.0$). Blind-IGT achieves near-oracle performance without prior knowledge of $\tau^*$.

| Method | $\tau$ Assumed | Error $\|\hat{\theta} - \theta^*\|_2$ | Std Dev |
|---|---|---|---|
| Blind-IGT | Estimated | **0.1238** | 0.0556 |
| Standard IGT | 1.0 (Misspec.) | 0.5027 | 0.0305 |
| Standard IGT | 2.0 (Oracle) | 0.1396 | 0.0567 |
| Standard IGT | 4.0 (Misspec.) | 1.0447 | 0.1242 |

### 6.3 The Necessity of Blind-IGT

We investigate the performance of standard IGT methods when the assumption of a known temperature is violated. We compare Blind-IGT (which estimates $\tau$) against Standard IGT (which assumes a fixed $\tau$) in a matrix game where $\tau^* = 2.0$. We consider the Oracle case ($\tau_{\mathrm{assumed}} = 2.0$) and two misspecified cases ($\tau_{\mathrm{assumed}} = 1.0$ and $4.0$). We use $N = 10000$ samples. The results are summarized in Table 1. When the temperature is misspecified, Standard IGT exhibits significant estimation errors. For instance, assuming $\tau = 1.0$ leads to an error almost 4 times higher than the Oracle. Crucially, Blind-IGT, without any prior knowledge of $\tau^*$, achieves an error of 0.1238, closely matching the Oracle performance. Oracle uses the same least-squares core as Blind-IGT and differs only in the final normalization. In our runs, Blind-IGT's norm-based scaling produces accuracy comparable to the Oracle, indicating that estimating $\tau$ via normalization is competitive with using the true $\tau^*$. This experiment strongly validates the core motivation of our framework.

### 6.4 Robustness to Unknown Dynamics in Markov Games.

We investigate the performance when the transition dynamics $P$ are unknown and must be estimated from data, relaxing the assumption made for Theorem 3.

We use the Markov Game setup ($|\mathcal{S}| = 8, m = n = 5, d = 6$). We apply Algorithm 2, estimating the transition dynamics $\hat{P}$ via Maximum Likelihood Estimation (with Laplace smoothing) from the observed trajectories. We compare this approach (Estimated-P) against an Oracle that uses the true dynamics (Known-P) during the reward recovery step.

Figure 4 shows the reward recovery error ($\|\hat{r} - r^*\|_\infty$). Crucially, the error for the Estimated-P approach resembles the Known-P oracle. The overhead introduced by estimating the dynamics is negligible, confirming the robustness of the framework in settings with unknown transitions.

## 7 CONCLUSION

This work establishes a new statistical foundation for Inverse Game Theory and multi-agent Inverse Reinforcement Learning by removing the restrictive assumption of known agent rationality. By rigorously addressing the bilinear coupling between rewards ($\theta$) and temperature ($\tau$), we resolved the fundamental scale ambiguity that previously precluded identification in entropy-regularized competitive games. Our analysis provides a complete characterization of identifiability through the statistically optimal NLS estimator, which achieves $\mathcal{O}(N^{-1/2})$ convergence rates.

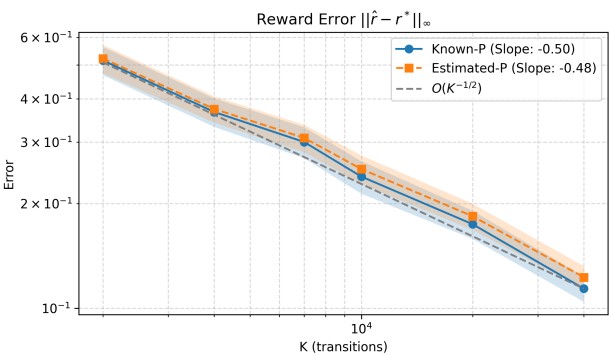

Figure 4: Robustness to Unknown Dynamics in Markov Games. The reward recovery error when estimating $P$ (Estimated-P) closely tracks the error when $P$ is known (Known-P). Both achieve near optimal $\mathcal{O}(K^{-1/2})$ rate (empirical slopes $-0.48$ and $-0.50$, respectively).

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

# A  Notations

We provide a summary of the key mathematical notation used in the paper and the supplementary materials.

Table 2: Summary of Notation

| Symbol | Description |
|---|---|
| **General** | |
| $\mathbb{R}^d$ | $d$-dimensional Euclidean space. |
| $\|\cdot\|_2, \|\cdot\|_1, \|\cdot\|_\infty$ | L2 (Euclidean), L1, and L-infinity norms for vectors. |
| $\|\cdot\|_{op}, \|\cdot\|_F$ | Operator (spectral) norm and Frobenius norm for matrices. |
| $\sigma_{\min}(X)$ | Minimum singular value of matrix $X$. |
| $X^\dagger$ | Moore-Penrose pseudoinverse of matrix $X$. |
| $\Delta(\mathcal{X})$ | Probability simplex over the set $\mathcal{X}$. |
| $\mathcal{H}(\pi)$ | Shannon entropy of the distribution $\pi$. |
| $A \lesssim B$ | $A \leq C \cdot B$ for some universal constant $C$. |
| **Game Setup** | |
| $\mathcal{A}, \mathcal{B}$ | Action spaces for Player 1 ($m = |\mathcal{A}|$) and Player 2 ($n = |\mathcal{B}|$). |
| $\mathcal{S}$ | State space (for Markov Games). |
| $Q$ | Payoff matrix (Matrix Games) or Q-function (Markov Games). |
| $r$ | Reward function (Markov Games). |
| $\tau$ | Temperature parameter (entropy regularization weight). |
| $\mu, \nu$ | Policies (mixed strategies) for Player 1 and Player 2. |
| $(\mu^*, \nu^*)$ | Quantal Response Equilibrium (QRE). |
| $\gamma$ | Discount factor (Markov Games). |
| $P$ | Transition dynamics (Markov Games). |
| **Parameterization and Estimation** | |
| $d$ | Dimension of the feature space. |
| $\phi$ | Feature map. $L = \sup \|\phi\|_2$. |
| $\theta^*$ | True reward (or Q-function) parameters. |
| $C, R$ | Normalization constants for $\|\theta^*\|_2$. |
| $N, K$ | Number of samples (Matrix Games), Total samples (Markov Games). |
| $\hat{\theta}, \hat{\tau}, \hat{Q}, \hat{r}$ | Estimated parameters and functions. |
| $\hat{\mu}, \hat{\nu}, \hat{P}$ | Estimated policies and dynamics. |
| $\xi$ | Soft-Min Gap (lower bound on QRE probabilities). |
| $C_{\min}$ | Coverage lower bound (Markov Games). |
| **System Matrices (Blind-IGT Formulation)** | |
| $A(\nu), B(\mu)$ | Feature matrices derived from linearized QRE constraints. |
| $c(\mu), d(\nu)$ | Log-ratio vectors. |
| $X(\mu, \nu)$ | Aggregated feature matrix $[A(\nu)^\top, B(\mu)^\top]^\top$. |
| $y(\mu, \nu)$ | Aggregated log-ratio vector $[c(\mu)^\top, d(\nu)^\top]^\top$. |
| $X^*, y^*$ | True system matrix and vector evaluated at QRE $(\mu^*, \nu^*)$. |
| $\hat{X}, \hat{y}$ | Estimated system matrix and vector evaluated at $(\hat{\mu}, \hat{\nu})$. |
| $\hat{\theta}_{LS}$ | Intermediate Least Squares solution (estimate of $\theta^*/\tau^*$). |
| $\theta'^*$ | True unnormalized parameters $\theta^*/\tau^*$. |

# B   Background and Derivations

This section provides foundational definitions, derivations, and external theorems used in the main paper and the subsequent proofs.

## B.1   Derivation of the Linearized QRE Constraints

We detail the derivation of the linearized QRE constraints (Eq. 4) from the QRE fixed-point equations (Eq. 2 and 3).

Recall the QRE definition for Player 1 (Eq. 2):

$$\mu^*(a) = \frac{\exp(Q(a,\cdot)\nu^*/\tau)}{\sum_{a'\in\mathcal{A}}\exp(Q(a',\cdot)\nu^*/\tau)}. \tag{17}$$

We select a reference action, conventionally action 1. We analyze the ratio of the probability of action $a$ to action 1:

$$\frac{\mu^*(a)}{\mu^*(1)} = \frac{\exp(Q(a,\cdot)\nu^*/\tau)/Z}{\exp(Q(1,\cdot)\nu^*/\tau)/Z} = \frac{\exp(Q(a,\cdot)\nu^*/\tau)}{\exp(Q(1,\cdot)\nu^*/\tau)}, \tag{18}$$

where $Z = \sum_{a'}\exp(Q(a',\cdot)\nu^*/\tau)$ is the partition function, which cancels out.

Taking the natural logarithm of both sides:

$$\log\left(\frac{\mu^*(a)}{\mu^*(1)}\right) = \log(\exp(Q(a,\cdot)\nu^*/\tau)) - \log(\exp(Q(1,\cdot)\nu^*/\tau)) \tag{19}$$

$$= \frac{1}{\tau}(Q(a,\cdot)\nu^* - Q(1,\cdot)\nu^*) \tag{20}$$

$$= \frac{1}{\tau}(Q(a,\cdot) - Q(1,\cdot))\nu^*. \tag{21}$$

Rearranging the terms yields the first set of linear constraints in Eq. 4:

$$(Q(a,\cdot) - Q(1,\cdot))\nu^* = \tau \cdot \log\left(\frac{\mu^*(a)}{\mu^*(1)}\right), \quad \forall a \in \mathcal{A} \setminus \{1\}. \tag{22}$$

Similarly, for Player 2 (Eq. 3), we analyze the ratio of $\nu^*(b)$ to $\nu^*(1)$:

$$\frac{\nu^*(b)}{\nu^*(1)} = \frac{\exp(-Q(\cdot,b)^\top\mu^*/\tau)}{\exp(-Q(\cdot,1)^\top\mu^*/\tau)}. \tag{23}$$

Taking the logarithm:

$$\log\left(\frac{\nu^*(b)}{\nu^*(1)}\right) = \frac{1}{\tau}(-Q(\cdot,b)^\top\mu^* - (-Q(\cdot,1)^\top\mu^*)) \tag{24}$$

$$= \frac{1}{\tau}(Q(\cdot,1)^\top - Q(\cdot,b)^\top)\mu^* \tag{25}$$

$$= \frac{1}{\tau}(Q(\cdot,1) - Q(\cdot,b))^\top\mu^*. \tag{26}$$

Rearranging the terms yields the second set of linear constraints in Eq. 4:

$$(Q(\cdot,1) - Q(\cdot,b))^\top\mu^* = \tau \cdot \log\left(\frac{\nu^*(b)}{\nu^*(1)}\right), \quad \forall b \in \mathcal{B} \setminus \{1\}. \tag{27}$$

This transformation converts the non-linear QRE fixed-point equations into a system of $m - 1 + n - 1$ linear equations with respect to the payoffs $Q$ and the temperature $\tau$.

### B.2 Key Definitions and Theorems from Matrix Analysis

We summarize essential concepts from matrix analysis used in our proofs.

**Definition 1** (Matrix Norms). Let $X \in \mathbb{R}^{m \times n}$.

- **Operator Norm (Spectral Norm),** $\|X\|_{op}$**:** The largest singular value of $X$, defined as $\|X\|_{op} = \sup_{\|v\|_2=1} \|Xv\|_2$.

- **Frobenius Norm,** $\|X\|_F$**:** The entry-wise L2 norm, defined as $\|X\|_F = \sqrt{\sum_{i,j} X_{ij}^2}$.

These norms satisfy the inequality $\|X\|_{op} \leq \|X\|_F$.

**Definition 2** (Moore-Penrose Pseudoinverse). For a matrix $X \in \mathbb{R}^{m \times n}$, the Moore-Penrose pseudoinverse $X^\dagger$ is the unique matrix satisfying the four Moore-Penrose conditions. If $X$ has full column rank (i.e., $\text{rank}(X) = n$), then $X^\dagger = (X^\top X)^{-1} X^\top$. In this case, $\|X^\dagger\|_{op} = 1/\sigma_{\min}(X)$.

**Theorem 1** (Weyl's Inequality for Singular Values). Let $A, B \in \mathbb{R}^{m \times n}$. Let $\sigma_i(X)$ denote the $i$-th largest singular value of $X$. Then for all $i$:

$$|\sigma_i(A) - \sigma_i(B)| \leq \|A - B\|_{op}. \tag{28}$$

In particular, for the minimum singular value (assuming $m \geq n$): $|\sigma_{\min}(A) - \sigma_{\min}(B)| \leq \|A - B\|_{op}$.

This theorem is crucial for establishing the stability of the matrix $X$ under small perturbations (Step 1 in the proof of Theorem 2).

### B.3 Key Theorems from Concentration Inequalities

We state the concentration inequality used to bound the policy estimation error.

**Theorem 2** (Bretagnolle-Huber-Carol Inequality). Let $\mu^*$ be a probability distribution over a finite set $\mathcal{A}$ of size $m$. Let $\hat{\mu}$ be the empirical distribution obtained from $N$ i.i.d. samples drawn from $\mu^*$. Then for any $\epsilon > 0$:

$$P(\|\hat{\mu} - \mu^*\|_1 > \epsilon) \leq 2^m e^{-N\epsilon^2/2}. \tag{29}$$

This theorem provides the finite-sample bound on the L1 distance between the estimated and true policies (Lemma 1).

## C Proofs for Matrix Games

This section provides the detailed proofs for the theoretical results presented in Section 4 concerning Blind-IGT in Matrix Games.

### C.1 Proof of Theorem 1 (Identifiability of Blind-IGT)

*Proof.* Let $X^* = X(\mu^*, \nu^*)$ and $y^* = y(\mu^*, \nu^*)$. The governing equation is the bilinear system $X^* \theta^* = \tau^* y^*$. We are given the constraints $\tau^* > 0$ and $\|\theta^*\|_2 = 1$ (assuming $C = 1$ WLOG).

**Proof of Sufficiency ($\Rightarrow$):** Assume the Rank Condition ($\text{rank}(X^*) = d$) and the Non-Uniformity Condition ($y^* \neq 0$) hold.

Since $X^*$ has full column rank, the Moore-Penrose pseudoinverse $(X^*)^\dagger = ((X^*)^\top X^*)^{-1}(X^*)^\top$ is well-defined and is a left inverse. By the definition of the QRE, the system $X^* \theta^* = \tau^* y^*$ is consistent. Applying the pseudoinverse yields:

$$\theta^* = (X^*)^\dagger (\tau^* y^*) = \tau^* (X^*)^\dagger y^*. \tag{30}$$

Let $\theta_{LS}^* = (X^*)^\dagger y^*$. This represents the least-squares solution assuming $\tau = 1$. Then $\theta^* = \tau^* \theta_{LS}^*$.

We must verify that $\theta_{LS}^* \neq 0$. Suppose, for contradiction, that $\theta_{LS}^* = 0$. Since the system is consistent, $y^*$ must be in the column space of $X^*$. The projection of $y^*$ onto this space is $P_X y^* = X^*(X^*)^\dagger y^*$. If $\theta_{LS}^* = 0$, then $P_X y^* = 0$. Since $y^*$ is in the column space, $P_X y^* = y^*$. Therefore, $y^* = 0$. This contradicts the Non-Uniformity Condition. Thus, $\theta_{LS}^* \neq 0$.

We now use the normalization constraint $\|\theta^*\|_2 = 1$:

$$\|\theta^*\|_2 = \|\tau^* \theta_{LS}^*\|_2 = |\tau^*| \|\theta_{LS}^*\|_2 = 1. \tag{31}$$

Since $\tau^* > 0$ (by definition of the regularized game) and $\|\theta_{LS}^*\|_2 > 0$ (as shown above), the temperature $\tau^*$ is uniquely identified as:

$$\tau^* = \frac{1}{\|\theta_{LS}^*\|_2}. \tag{32}$$

Consequently, the reward parameter vector $\theta^*$ is uniquely identified as:

$$\theta^* = \tau^* \theta_{LS}^* = \frac{\theta_{LS}^*}{\|\theta_{LS}^*\|_2}. \tag{33}$$

**Proof of Necessity ($\Leftarrow$):** We prove the contrapositive by considering the failure of each condition.

*Case 1: Rank Condition fails.* Suppose $\text{rank}(X^*) < d$. The null space $\mathcal{N}(X^*)$ is non-trivial. Let $v \in \mathcal{N}(X^*)$, $v \neq 0$. Let $(\theta^*, \tau^*)$ be the true solution, satisfying $X^* \theta^* = \tau^* y^*$ and $\|\theta^*\|_2 = 1$. Consider a perturbed vector $\theta' = \theta^* + \epsilon v$ for some $\epsilon \in \mathbb{R}$. We have $X^* \theta' = X^* \theta^* + \epsilon X^* v = \tau^* y^* + 0 = \tau^* y^*$. We normalize $\theta'$ to obtain a new candidate solution. Let $\bar{\theta}' = \theta'/\|\theta'\|_2$. Then $X^* \bar{\theta}' = (1/\|\theta'\|_2) X^* \theta' = (\tau^*/\|\theta'\|_2) y^*$. Defining $\bar{\tau}' = \tau^*/\|\theta'\|_2$. Since $\|\theta'\|_2$ depends on $\epsilon$, $\bar{\tau}'$ is generally different from $\tau^*$. We have found another pair $(\bar{\theta}', \bar{\tau}')$ that satisfies the constraints: $X^* \bar{\theta}' = \bar{\tau}' y^*$, $\|\bar{\theta}'\|_2 = 1$, and $\bar{\tau}' > 0$. For sufficiently small $\epsilon \neq 0$ such that $\theta' \neq 0$ (which is guaranteed since $\theta^* \neq 0$), we have $\bar{\theta}' \neq \theta^*$. Thus, the solution is not unique.

*Case 2: Non-Uniformity Condition fails.* Suppose $y^* = 0$. The bilinear system becomes $X^* \theta^* = \tau^* \cdot 0 = 0$. If the Rank Condition holds ($\text{rank}(X^*) = d$), the only solution to $X^* \theta^* = 0$ is $\theta^* = 0$. This violates the normalization constraint $\|\theta^*\|_2 = 1$. The problem is infeasible under the assumptions. If the Rank Condition fails ($\text{rank}(X^*) < d$), there are infinitely many solutions in the null space $\mathcal{N}(X^*)$. We can choose any $v \in \mathcal{N}(X^*)$ such that $\|v\|_2 = 1$. Then $(v, \tau)$ is a solution for any $\tau > 0$. $\theta^*$ is not uniquely identifiable.

Combining both cases, we conclude that both conditions are necessary for unique identification (or feasibility under the normalization constraint). $\qquad\square$

### C.2 Supporting Lemmas for Finite Sample Analysis

We present and prove the supporting lemmas required for the proof of Theorem 2. These lemmas characterize the concentration of the policy estimates and the resulting perturbations in the system matrices.

**Lemma 1** (Policy Estimation Error). *Given $N$ i.i.d. samples drawn from the QRE $(\mu^*, \nu^*)$, the empirical frequency estimators $(\hat{\mu}, \hat{\nu})$ satisfy, for any $\delta \in (0, 1)$, with probability at least $1 - \delta$:*

$$\|\hat{\mu} - \mu^*\|_1 \leq \epsilon_\mu, \quad \|\hat{\nu} - \nu^*\|_1 \leq \epsilon_\nu,$$

*where $\epsilon_\mu = \sqrt{\frac{2 \log(2 \cdot 2^m/\delta)}{N}}$ and $\epsilon_\nu = \sqrt{\frac{2 \log(2 \cdot 2^n/\delta)}{N}}$. We define the combined error rate $\epsilon_N = \epsilon_\mu + \epsilon_\nu = \mathcal{O}(\sqrt{(m + n + \log(1/\delta))/N})$.*

*Proof.* The empirical frequency estimator $\hat{\mu}$ is the maximum likelihood estimator for the multinomial distribution $\mu^*$. We utilize standard concentration bounds for the L1 distance (Total Variation distance) between the empirical distribution and the true distribution.

By the Bretagnolle-Huber-Carol inequality (Theorem 2 in Appendix B), for any $\epsilon > 0$:

$$P(\|\hat{\mu} - \mu^*\|_1 > \epsilon) \leq 2^m e^{-N\epsilon^2/2}. \tag{34}$$

Setting the RHS to $\delta/2$, we solve for $\epsilon_\mu$:

$$\frac{\delta}{2} = 2^m e^{-N\epsilon_\mu^2/2} \implies \log(2 \cdot 2^m/\delta) = N\epsilon_\mu^2/2 \implies \epsilon_\mu = \sqrt{\frac{2\log(2 \cdot 2^m/\delta)}{N}}. \tag{35}$$

Similarly, we define $\epsilon_\nu$ for $\hat{\nu}$. By a union bound, both bounds hold simultaneously with probability at least $1 - \delta$. The combined error rate $\epsilon_N$ scales as stated. $\qquad\square$

**Lemma 2** (Perturbation of X and y). *Let $L$ be the bound on $\|\phi\|_2$ (Assumption 1). Under Assumption 3, let $\xi > 0$ be the minimum probability. If $N$ is large enough such that the policy estimation errors satisfy $\epsilon_\mu < \xi/2$ and $\epsilon_\nu < \xi/2$, then with high probability (conditioned on the event of Lemma 1):*

$$\|\hat{X} - X^*\|_{op} \le C_X \epsilon_N, \quad \|\hat{y} - y^*\|_2 \le C_Y \epsilon_N.$$

*The constants are $C_X = 2L\sqrt{m+n}$ and $C_Y = \frac{\sqrt{8(m+n)}}{\xi}$.*

*Proof.* We analyze the perturbations of $X$ and $y$ separately.

**Bounding the perturbation of X.** The matrix $X$ is composed of $A(\nu)$ and $B(\mu)$. We analyze the perturbation of $A(\nu)$.

$$A(\hat{\nu}) - A(\nu^*) = A(\hat{\nu} - \nu^*). \tag{36}$$

This follows from the linearity of $A(\nu)$ with respect to $\nu$. The $(a-1)$-th row of $A(\nu)$ is $\sum_{b'} \nu(b')(\phi(a,b') - \phi(1,b'))^\top$. We bound the operator norm using the Frobenius norm:

$$\|A(\hat{\nu}) - A(\nu^*)\|_{op} \le \|A(\hat{\nu}) - A(\nu^*)\|_F = \sqrt{\sum_{a=2}^m \|(A(\hat{\nu}) - A(\nu^*))_{a-1}\|_2^2}. \tag{37}$$

We analyze a single row:

$$\|(A(\hat{\nu}) - A(\nu^*))_{a-1}\|_2 = \left\| \sum_{b' \in \mathcal{B}} (\hat{\nu}(b') - \nu^*(b'))(\phi(a,b') - \phi(1,b'))^\top \right\|_2 \tag{38}$$

$$\le \sum_{b' \in \mathcal{B}} |\hat{\nu}(b') - \nu^*(b')| \cdot \|\phi(a,b') - \phi(1,b')\|_2. \tag{39}$$

Since $\|\phi\|_2 \le L$, we have $\|\phi(a,b') - \phi(1,b')\|_2 \le 2L$.

$$\|(A(\hat{\nu}) - A(\nu^*))_{a-1}\|_2 \le 2L \sum_{b' \in \mathcal{B}} |\hat{\nu}(b') - \nu^*(b')| = 2L\|\hat{\nu} - \nu^*\|_1 \le 2L\epsilon_\nu. \tag{40}$$

Therefore, the Frobenius norm is bounded by:

$$\|A(\hat{\nu}) - A(\nu^*)\|_F \le \sqrt{(m-1)(2L\epsilon_\nu)^2} \le 2L\sqrt{m}\epsilon_\nu. \tag{41}$$

Similarly, $\|B(\hat{\mu}) - B(\mu^*)\|_F \le 2L\sqrt{n}\epsilon_\mu$.

The operator norm of the aggregated matrix $\hat{X} - X^*$ satisfies:

$$\|\hat{X} - X^*\|_{op} \le \|\hat{X} - X^*\|_F = \sqrt{\|A(\hat{\nu}) - A(\nu^*)\|_F^2 + \|B(\hat{\mu}) - B(\mu^*)\|_F^2} \tag{42}$$

$$\le \sqrt{4L^2 m \epsilon_\nu^2 + 4L^2 n \epsilon_\mu^2} = 2L\sqrt{m\epsilon_\nu^2 + n\epsilon_\mu^2}. \tag{43}$$

Since $\epsilon_\mu, \epsilon_\nu \le \epsilon_N$, we can bound this by $C_X \epsilon_N$, where $C_X = 2L\sqrt{m+n}$.

**Bounding the perturbation of y.** The vector $y$ is composed of $c(\mu)$ and $d(\nu)$. We analyze the perturbation of $c(\mu)$. $c(\mu)_{a-1} = \log(\mu(a)/\mu(1))$. The perturbation relies on the Lipschitz continuity of the logarithm function.

We use the assumption that $\epsilon_\mu < \xi/2$. Since $\|\hat{\mu} - \mu^*\|_\infty \le \|\hat{\mu} - \mu^*\|_1 \le \epsilon_\mu$, this ensures that the estimated probabilities are bounded away from zero:

$$\hat{\mu}(a) \ge \mu^*(a) - \|\hat{\mu} - \mu^*\|_\infty > \xi - \xi/2 = \xi/2. \tag{44}$$

The function $f(x) = \log(x)$ has a derivative $f'(x) = 1/x$. When $x \in [\xi/2, 1]$, the Lipschitz constant is $2/\xi$.

We analyze the difference in a single element of $c(\mu)$:

$$|c(\hat{\mu})_{a-1} - c(\mu^*)_{a-1}| = |\log(\hat{\mu}(a)/\hat{\mu}(1)) - \log(\mu^*(a)/\mu^*(1))| \tag{45}$$

$$= |\log(\hat{\mu}(a)) - \log(\mu^*(a)) - (\log(\hat{\mu}(1)) - \log(\mu^*(1)))|. \tag{46}$$

Using the Lipschitz property:

$$|\log(\hat{\mu}(a)) - \log(\mu^*(a))| \le \frac{2}{\xi}|\hat{\mu}(a) - \mu^*(a)|. \tag{47}$$

Therefore,

$$|c(\hat{\mu})_{a-1} - c(\mu^*)_{a-1}| \le \frac{2}{\xi}(|\hat{\mu}(a) - \mu^*(a)| + |\hat{\mu}(1) - \mu^*(1)|). \tag{48}$$

We bound the L2 norm of the difference:

$$\|c(\hat{\mu}) - c(\mu^*)\|_2^2 = \sum_{a=2}^m |c(\hat{\mu})_{a-1} - c(\mu^*)_{a-1}|^2 \tag{49}$$

$$\le \sum_{a=2}^m \left(\frac{2}{\xi}\right)^2 (|\hat{\mu}(a) - \mu^*(a)| + |\hat{\mu}(1) - \mu^*(1)|)^2 \tag{50}$$

$$\le \frac{8}{\xi^2} \sum_{a=2}^m (|\hat{\mu}(a) - \mu^*(a)|^2 + |\hat{\mu}(1) - \mu^*(1)|^2) \quad \text{(Since } (x+y)^2 \le 2(x^2 + y^2)) \tag{51}$$

$$= \frac{8}{\xi^2} \left(\sum_{a=2}^m |\hat{\mu}(a) - \mu^*(a)|^2 + (m-1)|\hat{\mu}(1) - \mu^*(1)|^2\right) \tag{52}$$

$$\le \frac{8m}{\xi^2}\|\hat{\mu} - \mu^*\|_2^2. \tag{53}$$

Since $\|\hat{\mu} - \mu^*\|_2 \le \|\hat{\mu} - \mu^*\|_1 \le \epsilon_\mu$, we have:

$$\|c(\hat{\mu}) - c(\mu^*)\|_2 \le \frac{\sqrt{8m}}{\xi}\epsilon_\mu. \tag{54}$$

Similarly, $\|d(\hat{\nu}) - d(\nu^*)\|_2 \le \frac{\sqrt{8n}}{\xi}\epsilon_\nu$.

The total perturbation in $y$ is:

$$\|\hat{y} - y^*\|_2 = \sqrt{\|c(\hat{\mu}) - c(\mu^*)\|_2^2 + \|d(\hat{\nu}) - d(\nu^*)\|_2^2} \tag{55}$$

$$\le \sqrt{\frac{8m}{\xi^2}\epsilon_\mu^2 + \frac{8n}{\xi^2}\epsilon_\nu^2} = \frac{\sqrt{8}}{\xi}\sqrt{m\epsilon_\mu^2 + n\epsilon_\nu^2}. \tag{56}$$

We can bound this by $C_Y \epsilon_N$, where $C_Y = \frac{\sqrt{8(m+n)}}{\xi}$. $\qquad\square$

**Lemma 3** (Stability of Normalization). *Let $v_0, v_1 \in \mathbb{R}^d$, $v_0 \ne 0$. If $\|v_1 - v_0\|_2 \le \epsilon$ and $\|v_0\|_2 \ge \delta > 0$. If $\epsilon < \delta/2$, then*

$$\left\|\frac{v_1}{\|v_1\|_2} - \frac{v_0}{\|v_0\|_2}\right\|_2 \le \frac{4\epsilon}{\delta}.$$

*Proof.* We analyze the norm of the difference between the normalized vectors:

$$\left\| \frac{v_1}{\|v_1\|_2} - \frac{v_0}{\|v_0\|_2} \right\|_2 = \left\| \frac{v_1\|v_0\|_2 - v_0\|v_1\|_2}{\|v_1\|_2\|v_0\|_2} \right\|_2. \tag{57}$$

Let $\Delta v = v_1 - v_0$ (where $\|\Delta v\|_2 \leq \epsilon$) and $\Delta_{\|v\|} = \|v_1\|_2 - \|v_0\|_2$. By the reverse triangle inequality, $|\Delta_{\|v\|}| \leq \|v_1 - v_0\|_2 \leq \epsilon$.

We analyze the numerator:

$$\|v_1\|v_0\|_2 - v_0\|v_1\|_2\| = \|(v_0 + \Delta v)\|v_0\|_2 - v_0(\|v_0\|_2 + \Delta_{\|v\|})\| \tag{58}$$

$$= \|v_0\|v_0\|_2 + \Delta v\|v_0\|_2 - v_0\|v_0\|_2 - v_0\Delta_{\|v\|}\| \tag{59}$$

$$= \|\Delta v\|v_0\|_2 - v_0\Delta_{\|v\|}\|. \tag{60}$$

Using the triangle inequality:

$$\|\Delta v\|v_0\|_2 - v_0\Delta_{\|v\|}\| \leq \|\Delta v\|v_0\|_2\| + \|v_0\Delta_{\|v\|}\| \tag{61}$$

$$= \|\Delta v\|_2\|v_0\|_2 + |\Delta_{\|v\|}|\|v_0\|_2 \tag{62}$$

$$\leq \epsilon\|v_0\|_2 + \epsilon\|v_0\|_2 = 2\epsilon\|v_0\|_2. \tag{63}$$

We analyze the denominator: $\|v_1\|_2\|v_0\|_2$. We need a lower bound for $\|v_1\|_2$.

$$\|v_1\|_2 \geq \|v_0\|_2 - \|v_1 - v_0\|_2 \geq \delta - \epsilon. \tag{64}$$

Since we assumed $\epsilon < \delta/2$, we have $\|v_1\|_2 > \delta/2$.

Combining the bounds for the numerator and denominator:

$$\left\| \frac{v_1}{\|v_1\|_2} - \frac{v_0}{\|v_0\|_2} \right\|_2 \leq \frac{2\epsilon\|v_0\|_2}{(\delta/2)\|v_0\|_2} = \frac{4\epsilon}{\delta}. \tag{65}$$

$\square$

## C.3 Proof of Theorem 2 (Finite Sample Bounds)

We now combine the supporting lemmas to prove the main finite-sample result. We assume $C = 1$ without loss of generality.

*Proof.* Let $\sigma^* = \sigma_{\min}(X^*)$. Since the Rank Condition holds (Theorem 1), $\sigma^* > 0$. Let $\epsilon_N$ be the policy estimation error rate from Lemma 1. Let $C_X, C_Y$ be the perturbation constants from Lemma 2.

The proof proceeds in three main steps: (1) Ensuring the stability of the estimated system, (2) Bounding the error of the intermediate LS solution, and (3) Bounding the error of the normalized solution and the temperature estimate.

**Stability of the Estimated System.** We need to ensure that the estimated matrix $\hat{X}$ remains full rank and well-conditioned, provided $N$ is sufficiently large. We use Weyl's inequality (Theorem 1 in Appendix B) for the perturbation of singular values:

$$\sigma_{\min}(\hat{X}) \geq \sigma_{\min}(X^*) - \|\hat{X} - X^*\|_{op} = \sigma^* - \|\hat{X} - X^*\|_{op}. \tag{66}$$

We require $\|\hat{X} - X^*\|_{op} < \sigma^*/2$ to guarantee $\sigma_{\min}(\hat{X}) > \sigma^*/2$. By Lemma 2, this holds if $C_X\epsilon_N < \sigma^*/2$. This defines the primary condition for "sufficiently large $N$". We also require $N$ large enough such that the conditions of Lemma 2 (i.e., $\epsilon_\mu, \epsilon_\nu < \xi/2$) hold.

**Error in the Least Squares Solution.** Let $\theta'^* = \theta^*/\tau^*$. This is the true solution to the unnormalized system $X^*\theta'^* = y^*$. The NLS algorithm computes the empirical LS solution $\hat{\theta}_{LS} = (\hat{X}^\top\hat{X})^{-1}\hat{X}^\top\hat{y}$. We analyze the error $\|\hat{\theta}_{LS} - \theta'^*\|_2$.

We utilize standard perturbation bounds for the least squares solution when the true system is consistent (zero residual). Since $\hat{X}$ is full rank (Step 1), $\hat{\theta}_{LS} = \hat{X}^\dagger \hat{y}$. The error bound is given by (e.g., Theorem 20.1 in Higham (2002), specialized for zero residual):

$$\|\hat{\theta}_{LS} - \theta'^*\|_2 \leq \|\hat{X}^\dagger\|_2 (\|\hat{X} - X^*\|_{op} \|\theta'^*\|_2 + \|\hat{y} - y^*\|_2) + \mathcal{O}(\epsilon_N^2). \tag{67}$$

Since $\sigma_{\min}(\hat{X}) \geq \sigma^*/2$, we have $\|\hat{X}^\dagger\|_2 = 1/\sigma_{\min}(\hat{X}) \leq 2/\sigma^*$. We substitute the bounds from Lemma 2 and note that $\|\theta'^*\|_2 = \|\theta^*\|_2/\tau^* = 1/\tau^*$ (since $C = 1$).

$$\|\hat{\theta}_{LS} - \theta'^*\|_2 \leq \frac{2}{\sigma^*} \left( (C_X \epsilon_N) \frac{1}{\tau^*} + C_Y \epsilon_N \right) \tag{68}$$

$$= \left( \frac{2C_X}{\sigma^* \tau^*} + \frac{2C_Y}{\sigma^*} \right) \epsilon_N =: C_{LS} \epsilon_N. \tag{69}$$

This establishes that the intermediate LS solution converges at the optimal rate $\mathcal{O}(N^{-1/2})$.

**Error in Normalized Solution and Temperature.** We now analyze the effect of the normalization steps.

*Error in $\hat{\theta}$.* We apply Lemma 3 (Stability of Normalization) with $v_1 = \hat{\theta}_{LS}$ and $v_0 = \theta'^*$. The error is $\epsilon = \|\hat{\theta}_{LS} - \theta'^*\|_2 \leq C_{LS} \epsilon_N$. The norm is $\delta = \|\theta'^*\|_2 = 1/\tau^*$. We require $\epsilon < \delta/2$, i.e., $C_{LS} \epsilon_N < 1/(2\tau^*)$. This holds for sufficiently large $N$ since $\epsilon_N \to 0$.

The error in the normalized estimator $\hat{\theta} = \hat{\theta}_{LS}/\|\hat{\theta}_{LS}\|_2$ is:

$$\|\hat{\theta} - \theta^*\|_2 = \left\| \frac{\hat{\theta}_{LS}}{\|\hat{\theta}_{LS}\|_2} - \frac{\theta'^*}{\|\theta'^*\|_2} \right\|_2 \tag{70}$$

$$\leq \frac{4\|\hat{\theta}_{LS} - \theta'^*\|_2}{\|\theta'^*\|_2} = \frac{4C_{LS}\epsilon_N}{1/\tau^*} = 4\tau^* C_{LS} \epsilon_N. \tag{71}$$

We define $C_\theta = 4\tau^* C_{LS}$.

*Error in $\hat{\tau}$.* The temperature estimate is $\hat{\tau} = 1/\|\hat{\theta}_{LS}\|_2$ (since $C = 1$). The true temperature is $\tau^* = 1/\|\theta'^*\|_2$. We analyze the error:

$$|\hat{\tau} - \tau^*| = \left| \frac{1}{\|\hat{\theta}_{LS}\|_2} - \frac{1}{\|\theta'^*\|_2} \right| = \frac{|\|\theta'^*\|_2 - \|\hat{\theta}_{LS}\|_2|}{\|\hat{\theta}_{LS}\|_2 \|\theta'^*\|_2}. \tag{72}$$

The numerator is bounded by the reverse triangle inequality: $|\|\theta'^*\|_2 - \|\hat{\theta}_{LS}\|_2| \leq \|\hat{\theta}_{LS} - \theta'^*\|_2 \leq C_{LS}\epsilon_N$. The denominator is bounded below. Since $N$ is large enough (as established above), $\|\hat{\theta}_{LS}\|_2 \geq \|\theta'^*\|_2/2$.

$$|\hat{\tau} - \tau^*| \leq \frac{C_{LS}\epsilon_N}{(\|\theta'^*\|_2/2)\|\theta'^*\|_2} = \frac{2C_{LS}\epsilon_N}{\|\theta'^*\|_2^2} \tag{73}$$

$$= 2(\tau^*)^2 C_{LS} \epsilon_N. \tag{74}$$

We define $C_\tau = 2(\tau^*)^2 C_{LS}$.

Since $\epsilon_N = \mathcal{O}(\sqrt{(m + n + \log(1/\delta))/N})$, this completes the proof, establishing the desired parametric rate $\mathcal{O}(N^{-1/2})$ for both $\hat{\theta}$ and $\hat{\tau}$. $\qquad\square$

## C.4 Proof of Proposition 1 (Partial Identification)

*Proof.* The goal is to ensure that the confidence set $\hat{\Theta}_N$ covers the identified set $\Theta^*$ with probability at least $1 - \delta$. This requires that for all $(\theta^*, \tau^*) \in \Theta^*$, the condition $\|\hat{X}\theta^* - \tau^*\hat{y}\|_2^2 \leq \kappa_N$ holds with high probability.

Let $(\theta^*, \tau^*) \in \Theta^*$. By definition, $X^*\theta^* = \tau^*y^*$, $\|\theta^*\|_2 = C$, and $\tau^* > 0$. We analyze the empirical residual:

$$\|\hat{X}\theta^* - \tau^*\hat{y}\|_2 = \|\hat{X}\theta^* - X^*\theta^* + X^*\theta^* - \tau^*\hat{y}\|_2$$

$$= \|\hat{X}\theta^* - X^*\theta^* + \tau^*y^* - \tau^*\hat{y}\|_2 \quad \text{(Since } X^*\theta^* = \tau^*y^*)$$

$$= \|(\hat{X} - X^*)\theta^* - \tau^*(\hat{y} - y^*)\|_2.$$

Using the triangle inequality and the definition of the operator norm:

$$\|\hat{X}\theta^* - \tau^*\hat{y}\|_2 \leq \|(\hat{X} - X^*)\theta^*\|_2 + \|\tau^*(\hat{y} - y^*)\|_2$$
$$\leq \|\hat{X} - X^*\|_{op}\|\theta^*\|_2 + \tau^*\|\hat{y} - y^*\|_2.$$

We now substitute the bounds established in the supporting lemmas. Let $\epsilon_N$ be the convergence rate of the policy estimation (Lemma 1). From Lemma 2, we know that with probability at least $1 - \delta$ (assuming $N$ is sufficiently large such that the conditions of the lemma hold):

$$\|\hat{X} - X^*\|_{op} \leq C_X \epsilon_N, \quad \|\hat{y} - y^*\|_2 \leq C_Y \epsilon_N.$$

Substituting these bounds and the normalization constraint $\|\theta^*\|_2 = C$:

$$\|\hat{X}\theta^* - \tau^*\hat{y}\|_2 \leq (C_X \epsilon_N) \cdot C + \tau^* \cdot (C_Y \epsilon_N)$$
$$= (C_X C + C_Y \tau^*)\epsilon_N.$$

By Assumption 4, we have $\tau^* \leq \tau_{max}$. Therefore, the residual is uniformly bounded over $\Theta^*$:

$$\sup_{(\theta^*, \tau^*) \in \Theta^*} \|\hat{X}\theta^* - \tau^*\hat{y}\|_2 \leq (C_X C + C_Y \tau_{max})\epsilon_N.$$

Squaring this bound gives the required threshold for $\kappa_N$:

$$\kappa_N \geq (C_X C + C_Y \tau_{max})^2 \cdot \epsilon_N^2.$$

By choosing $\kappa_N$ according to this inequality, we ensure that $\Theta^* \subseteq \hat{\Theta}_N$ with probability at least $1 - \delta$. □

## D  Proofs for Markov Games

This section provides the detailed proofs for the theoretical results presented in Section 5 concerning Blind-IGT in Markov Games. We assume the dynamics $P$ are known, as stated in Theorem 3. We assume Assumption 3 holds uniformly for the QRE policies at every state $s \in \mathcal{S}$, i.e., $\min_{s,a,b}\{\mu^*(a|s), \nu^*(b|s)\} \geq \xi$.

### D.1  Proof of Proposition 2 (Identifiability in MGs)

*Proof.* The aggregated system of equations is $X^*\theta^* = \tau^*y^*$, where $X^*$ is formed by stacking the local matrices $X(s; \mu^*, \nu^*)$ vertically, and $y^*$ by stacking the local vectors $y(s; \mu^*, \nu^*)$ vertically. The constraints are $\|\theta^*\|_2 = 1$ (assuming $R = 1$) and $\tau^* > 0$.

The mathematical structure of this aggregated bilinear system is identical to the matrix game formulation analyzed in Theorem 1. Therefore, the necessary and sufficient conditions for unique identification directly translate. The pair $(\theta^*, \tau^*)$ is uniquely identifiable if and only if:

1. The aggregated matrix $X^*$ has full column rank (rank $d$).

2. The aggregated log-ratio vector $y^* \neq 0$.

The proof follows exactly the steps outlined in Appendix C.1, applied to the aggregated matrices $X^*$ and $y^*$. □

### D.2  Proof of Theorem 3 (Sample Complexity for MGs)

The proof builds upon the matrix game analysis (Appendix C.3) and incorporates the complexities of dynamic programming, specifically the error propagation during reward recovery.

*Proof.* Let $K$ be the total number of samples. We assume $R = 1$ WLOG.

**Uniform Policy Estimation Error.** We first establish the uniform convergence of the QRE policy estimates $(\hat{\mu}(s), \hat{\nu}(s))$. We observe samples drawn from the stationary distribution $d^*(s)$ induced by the QRE. Under Assumption 6 $(d^*(s) \geq C_{\min} > 0)$, the expected number of visits to state $s$ is approximately $K \cdot d^*(s) \geq KC_{\min}$.

We utilize concentration results for empirical estimates in Markov chains (or assume access to a generative model allowing sampling from $d^*$). This ensures that the empirical estimators converge uniformly across all states $s \in \mathcal{S}$. By adapting the results from Lemma 1 and applying a union bound over all states, we obtain that with probability at least $1 - \delta$:

$$\sup_s(\|\hat{\mu}(s) - \mu^*(s)\|_1 + \|\hat{\nu}(s) - \nu^*(s)\|_1) = \mathcal{O}\left(\sqrt{\frac{(m+n)\log(|\mathcal{S}|mn/\delta)}{KC_{\min}}}\right). \tag{75}$$

We denote this uniform error rate as $\epsilon_K$.

**Estimation of $\theta^*$ and $\tau^*$.** The estimation of $\theta^*$ and $\tau^*$ relies on the aggregated system $\hat{X}\hat{\theta}_{LS} = \hat{y}$. The aggregated matrices $\hat{X}$ and $\hat{y}$ are formed by stacking the local estimates $\hat{X}(s)$ and $\hat{y}(s)$.

We analyze the perturbation of the aggregated system.

$$\|\hat{X} - X^*\|_{op} \leq \|\hat{X} - X^*\|_F = \sqrt{\sum_s \|\hat{X}(s) - X^*(s)\|_F^2}. \tag{76}$$

Using the analysis from Lemma 2, the local perturbation $\|\hat{X}(s) - X^*(s)\|_F$ is bounded by $C_X(s)\epsilon_K(s)$, where $\epsilon_K(s)$ is the local policy error at state $s$. Since we established uniform convergence $\epsilon_K$, and $C_X$ depends on $L, m, n$ (which are uniform across states), we have:

$$\|\hat{X} - X^*\|_{op} \leq \sqrt{|\mathcal{S}|C_X^2\epsilon_K^2} = \sqrt{|\mathcal{S}|}C_X\epsilon_K. \tag{77}$$

Similarly, $\|\hat{y} - y^*\|_2 \leq \sqrt{|\mathcal{S}|}C_Y\epsilon_K$.

We now apply the analysis of Theorem 2 (Appendix C.3) to the aggregated system. Provided $K$ is sufficiently large such that $\sigma_{\min}(\hat{X})$ is well-conditioned (relative to $\sigma_{\min}(X^*)$), the error in the LS solution $\hat{\theta}_{LS}$ compared to $\theta'^* = \theta^*/\tau^*$ is:

$$\|\hat{\theta}_{LS} - \theta'^*\|_2 = \mathcal{O}(\sqrt{|\mathcal{S}|}\epsilon_K). \tag{78}$$

Applying the normalization stability (Lemma 3), we obtain the bounds for the parameters:

$$\|\hat{\theta} - \theta^*\|_2 = \mathcal{O}(\sqrt{|\mathcal{S}|}\epsilon_K), \quad |\hat{\tau} - \tau^*| = \mathcal{O}(\sqrt{|\mathcal{S}|}\epsilon_K). \tag{79}$$

This confirms the rates stated in the theorem for $\hat{\theta}$ and $\hat{\tau}$ (noting the definition of $\epsilon_K$).

**Error Propagation in Reward Recovery.** The crucial step is analyzing the error in the recovered rewards $\hat{r}$. We assume the true dynamics $P$ are used for recovery (as stated in the theorem assumptions).

$$\hat{r}(s, a, b) = \hat{Q}(s, a, b) - \gamma\mathbb{E}_{s' \sim P(\cdot|s,a,b)}[\hat{V}(s')]. \tag{80}$$

The error decomposition is:

$$\hat{r} - r^* = (\hat{Q} - Q^*) - \gamma P(\hat{V} - V^*). \tag{81}$$

Where $P(\hat{V} - V^*)$ denotes the vector of expected value differences.

*Bounding Q-function Error.*

$$\|\hat{Q} - Q^*\|_\infty = \sup_{s,a,b}|\langle\phi(s, a, b), \hat{\theta} - \theta^*\rangle| \leq L\|\hat{\theta} - \theta^*\|_2 = \mathcal{O}(\sqrt{|\mathcal{S}|}\epsilon_K). \tag{82}$$

*Bounding V-function Error.* We need to bound $\|\hat{V} - V^*\|_\infty$. The V-function at a state $s$ is defined as a function of the local components:

$$V(s) = F(Q(s), \mu(s), \nu(s), \tau) = \mu^\top Q \nu + \tau(\mathcal{H}(\mu) - \mathcal{H}(\nu)). \tag{83}$$

The estimated V-function $\hat{V}(s)$ is calculated using the estimated components $(\hat{Q}(s), \hat{\mu}(s), \hat{\nu}(s), \hat{\tau})$.

We analyze the Lipschitz continuity of the function $F$. Provided the policies are bounded away from zero (by $\xi > 0$, Assumption 3), $F$ is Lipschitz continuous with respect to its arguments.

We analyze the sensitivity of $F$ to its inputs. Let $\Delta V = \hat{V}(s) - V^*(s)$.

$$\Delta V = (\hat{\mu}^\top \hat{Q} \hat{\nu} - \mu^{*\top} Q^* \nu^*) + (\hat{\tau}\mathcal{H}(\hat{\mu}) - \tau^*\mathcal{H}(\mu^*)) - (\hat{\tau}\mathcal{H}(\hat{\nu}) - \tau^*\mathcal{H}(\nu^*)).$$

We analyze the first term (the expected payoff):

$$\begin{aligned}
\hat{\mu}^\top \hat{Q} \hat{\nu} - \mu^{*\top} Q^* \nu^* &= \hat{\mu}^\top (\hat{Q} - Q^*)\hat{\nu} + \hat{\mu}^\top Q^* \hat{\nu} - \mu^{*\top} Q^* \nu^* \\
&= \hat{\mu}^\top (\hat{Q} - Q^*)\hat{\nu} + (\hat{\mu} - \mu^*)^\top Q^* \hat{\nu} + \mu^{*\top} Q^* (\hat{\nu} - \nu^*).
\end{aligned}$$

The magnitude is bounded by:

$$\begin{aligned}
|\hat{\mu}^\top \hat{Q} \hat{\nu} - \mu^{*\top} Q^* \nu^*| &\leq \|\hat{Q} - Q^*\|_\infty + \|Q^*\|_\infty (\|\hat{\mu} - \mu^*\|_1 + \|\hat{\nu} - \nu^*\|_1) \\
&\leq \|\hat{Q} - Q^*\|_\infty + L \cdot \epsilon_K. \quad (\text{Since } \|Q^*\|_\infty \leq L, R = 1)
\end{aligned}$$

We analyze the entropy terms. The entropy function $\mathcal{H}(\pi)$ is Lipschitz continuous when $\pi_i \geq \xi/2$. The Lipschitz constant is $L_\mathcal{H} \approx \log(1/\xi)$.

$$\begin{aligned}
|\hat{\tau}\mathcal{H}(\hat{\mu}) - \tau^*\mathcal{H}(\mu^*)| &= |\hat{\tau}(\mathcal{H}(\hat{\mu}) - \mathcal{H}(\mu^*)) + (\hat{\tau} - \tau^*)\mathcal{H}(\mu^*)| \\
&\leq |\hat{\tau}|L_\mathcal{H}\|\hat{\mu} - \mu^*\|_1 + |\hat{\tau} - \tau^*|\log(m).
\end{aligned}$$

Combining these bounds, we see that $\|\hat{V} - V^*\|_\infty$ is bounded by a linear combination of the errors in the estimated components:

$$\|\hat{V} - V^*\|_\infty \leq L_Q\|\hat{Q} - Q^*\|_\infty + L_\tau|\hat{\tau} - \tau^*| + L_{\mu,\nu}\epsilon_K.$$

The errors in $\hat{Q}$ and $\hat{\tau}$ are $\mathcal{O}(\sqrt{|\mathcal{S}|}\epsilon_K)$. Since this dominates $\mathcal{O}(\epsilon_K)$, we conclude that:

$$\|\hat{V} - V^*\|_\infty = \mathcal{O}(\sqrt{|\mathcal{S}|}\epsilon_K). \tag{84}$$

**Final Reward Error Bound.** We combine the errors to bound the reward recovery error:

$$\|\hat{r} - r^*\|_\infty \leq \|\hat{Q} - Q^*\|_\infty + \gamma\|P(\hat{V} - V^*)\|_\infty.$$

Since $P$ is a stochastic transition matrix, applying $P$ is an averaging operation, so $\|P(\hat{V}-V^*)\|_\infty \leq \|\hat{V}-V^*\|_\infty$.

$$\begin{aligned}
\|\hat{r} - r^*\|_\infty &\leq \|\hat{Q} - Q^*\|_\infty + \gamma\|\hat{V} - V^*\|_\infty \\
&= \mathcal{O}(\sqrt{|\mathcal{S}|}\epsilon_K).
\end{aligned}$$

Substituting the definition of $\epsilon_K$ completes the proof, confirming the $\mathcal{O}(K^{-1/2})$ rate for reward recovery in Markov games under the assumption of known dynamics. $\qquad\square$

# E   Experimental Details and Additional Results

## E.1   Experimental Setup Details

This section provides further details on the experimental setup described in Section 6.

**Matrix Games Generation.** For the matrix game experiments (Sections 6.2, 6.3), we set the dimensions $m = 10, n = 10$ and the feature dimension $d = 5$. The features $\phi(a, b) \in \mathbb{R}^d$ were generated by sampling each element independently from a standard normal distribution $\mathcal{N}(0, 1)$. The true reward parameter $\theta^* \in \mathbb{R}^d$ was also sampled from $\mathcal{N}(0, 1)$ and then normalized such that $\|\theta^*\|_2 = 1$ (Normalization constant $C = 1$), unless otherwise specified (e.g., Section 4.5). The true temperature was fixed at $\tau^* = 2.0$. The payoff matrix $Q^*$ was computed as $Q^*(a, b) = \langle \phi(a, b), \theta^* \rangle$. The QRE $(\mu^*, \nu^*)$ was computed by solving the fixed-point equations (Eq. 2, 3) using iterative updates until convergence (tolerance $10^{-9}$), falling back to a numerical root-finding solver if iterations failed. Samples were drawn directly from the computed QRE.

**Markov Games Generation.** For the Markov game experiments (Sections 6.2, 6.4), we set $|\mathcal{S}| = 8, m = 5, n = 5, d = 6$, and $\gamma = 0.9$. We adopted the Linear Q-function parameterization (Assumption 5). Features $\phi(s, a, b) \in \mathbb{R}^d$ were sampled from $\mathcal{N}(0, 1)$. The true parameter $\theta^*$ was sampled from $\mathcal{N}(0, 1)$ and normalized to $\|\theta^*\|_2 = 1$ (Normalization constant $R = 1$). The true temperature was $\tau^* = 1.5$. The QRE Q-function $Q^*$ was defined as $Q^*(s, a, b) = \langle \phi(s, a, b), \theta^* \rangle$. The QRE policies $(\mu^*, \nu^*)$ were computed from $Q^*$ using the logit response equations (using the same solver approach as Matrix Games). The V-function $V^*$ was computed using Eq. 6.

The transition dynamics $P(s'|s, a, b)$ were generated randomly by sampling from a Dirichlet distribution $\text{Dir}(1)$ for each $(s, a, b)$. The true reward function $r^*$ was then computed by inverting the Bellman equation (Eq. 5): $r^*(s, a, b) = Q^*(s, a, b) - \gamma \mathbb{E}_{s' \sim P(\cdot|s,a,b)}[V^*(s')]$.

Samples were generated using a generative model approach, where a fixed number of samples ($N_{\text{per\_state}}$) were drawn from the QRE policies $(\mu^*, \nu^*)$ independently at each state $s \in \mathcal{S}$. This ensures uniform coverage across the state space. $K = |\mathcal{S}| \cdot N_{\text{per\_state}}$ denotes the total number of state-action-next state tuples collected.

## E.2   Robustness to Feature Misspecification

A critical assumption in IGT/IRL is that the feature map $\phi$ (Assumption 1) perfectly captures the relevant factors influencing the agents' utilities. We analyze the robustness of Blind-IGT when this assumption is violated, specifically when the feature map used for estimation is incomplete.

### E.2.1   Setup

We use the Matrix Game setup ($m = n = 10, d = 5, \tau^* = 2.0$) but provide the NLS estimator with only $d_{est} = 4$ features (omitting one true feature). We measure (1) the Directional Error (measured as $1 - $ cosine similarity) between the estimated $\hat{\theta}$ (in the $d_{est}$ space) and the parameters that best approximate the true Q-values within the restricted feature space (obtained via least-squares projection, i.e., $\arg\min_\theta \|\Phi_{est}\theta - Q^*\|_2^2$), and (2) the Behavioral Error (Total Variation distance) between the policies induced by the estimated parameters and the true QRE policies.

### E.2.2   Results

Figure 5 shows the results. Both the directional error and the behavioral error decrease as the sample size $N$ increases. While the errors are naturally higher than in the perfectly specified case (Figure 1), the behavioral error remains relatively small. This indicates that even when the underlying utility model is misspecified, Blind-IGT can still recover parameters that accurately model the observed behavior.

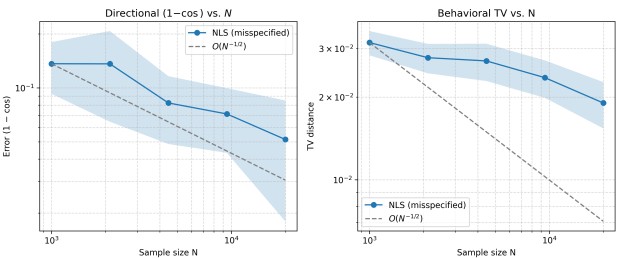

Figure 5: Robustness to Feature Misspecification.

## F    Algorithms

We present the full pseudocode for the algorithms proposed in this work.

---

**Algorithm 1:** Normalized Least Squares (NLS) for Matrix Games

---

**Input:** Data $\{(a^k, b^k)\}_{k=1}^N$, features $\phi$, Norm $C$
**Output:** Estimated parameters $(\hat{\theta}, \hat{\tau})$

   /* 1.  Estimate QRE from observed frequencies                    */
**1** $(\hat{\mu}, \hat{\nu}) \leftarrow$ frequency estimators from data

   /* 2.  Construct regression matrices                              */
**2** $\hat{X} \leftarrow X(\hat{\mu}, \hat{\nu})$
**3** $\hat{y} \leftarrow y(\hat{\mu}, \hat{\nu})$

   /* 3.  Solve least squares problem                                 */
**4** $\hat{\theta}_{LS} \leftarrow (\hat{X}^\top \hat{X})^{-1} \hat{X}^\top \hat{y}$

   /* 4.  Recover temperature and reward parameters            */
**5** $\hat{\tau} \leftarrow \dfrac{C}{\|\hat{\theta}_{LS}\|_2}$
**6** $\hat{\theta} \leftarrow \hat{\tau} \cdot \hat{\theta}_{LS}$

**7 return** $(\hat{\theta}, \hat{\tau})$

---

---

**Algorithm 2:** Normalized Least Squares for Markov Games

---

**Input:** Trajectories, features $\phi$, Norm $R$, Dynamics $P$ (or estimate)
**Output:** $(\hat{\theta}, \hat{\tau}, \hat{r})$

   /* 1.  Estimate policies from trajectories                      */
**1** $(\hat{\mu}(s), \hat{\nu}(s)) \leftarrow$ frequency estimators for all $s \in \mathcal{S}$

   /* 2.  Estimate dynamics if unknown                           */
**2 if** $P$ *unknown* **then**
**3**    |  $\hat{P} \leftarrow$ MLE from observed transitions
**4 end**

   /* 3.  Construct aggregated system                           */
**5** $\hat{X} \leftarrow$ stack $X(s; \hat{\mu}, \hat{\nu})$ for all $s$
**6** $\hat{y} \leftarrow$ stack $y(s; \hat{\mu}, \hat{\nu})$ for all $s$

   /* 4.  Apply NLS procedure                                  */
**7** $\hat{\theta}_{LS} \leftarrow (\hat{X}^\top \hat{X})^{-1} \hat{X}^\top \hat{y}$
**8** $\hat{\tau} \leftarrow R / \|\hat{\theta}_{LS}\|_2$
**9** $\hat{\theta} \leftarrow \hat{\tau} \cdot \hat{\theta}_{LS}$

   /* 5.  Recover rewards via Bellman equation               */
**10** $\hat{Q}(s, a, b) \leftarrow \langle \phi(s, a, b), \hat{\theta} \rangle$ for all $(s, a, b)$
**11** Compute $\hat{V}(s)$ from $\hat{Q}, \hat{\mu}, \hat{\nu}, \hat{\tau}$
**12** $\hat{r}(s, a, b) \leftarrow \hat{Q}(s, a, b) - \gamma \mathbb{E}_{s'}[\hat{V}(s')]$

**13 return** $(\hat{\theta}, \hat{\tau}, \hat{r})$

---

