# OpenReview forum: "Blind Inverse Game Theory: Jointly Decoding Rewards and Rationality in Entropy-Regularized Competitive Games"
_TMLR — Withdrawn by Authors_

### Review · Reviewer_6P9h · 2026-03-13

**Summary Of Contributions:**

This work proposes an algorithm for learning the underlying payoffs/rewards and rationality parameter ($\tau$) of a zero-sum bimatrix game or Markov game given knowledge of the norm of the underlying payoff parameters ($\theta$) and observations of play. The authors prove a $\mathcal{O}(N^{-1/2})$ convergence rate to the true parameters given N samples from the underlying quantal response equilibrium. They analyze the cases for both when the parameters are strongly identifiable (uniquely determined) and partial identifiable from the data. Experiments empirically support their theoretical derivations.

**Audience:**

Yes

**Audience Explanation:**

I think researchers would be interested to know that the parameters of zero-sum games can be learned at a $\mathcal{N}^{-1/2}$ rate given $N$ samples of play from a supposed QRE (and the norm of the parameters). QREs are a popular model of bounded rationality solutions to games and the authors address both zero-sum matrix and Markov games.

**Broader Impact Concerns:**

No concerns.

**Claims And Evidence:**

Yes

**Claims Explanation:**

I say yes, but with a caveat. The paper claims in several places to "resolve the scale ambiguity" problem of estimating $\theta$ and $\tau$. The proposed approach relies on Assumption 2 which assumes the scale of $\theta$ so that the direction of $\theta$ and scale of $\tau$ can then be estimated. This is stated in contrast to works that assume the scale of $\tau$. One, I don't think the authors can claim to "resolve" the scale ambiguity problem if the key is *assuming the scale* of $\theta$. Two, I don't see a difference mathematically between assuming the scale of $\theta$ and assuming the scale of $\tau$. They are interchangeable due to symmetry in the constraints. In that sense, can't prior work claim to have resolved the scale ambiguity problem in the same way? My criticism here is mostly regarding the portrayal of this contribution.

Aside from resolution of the scale ambiguity problem, I believe statistical rates for estimation of the game parameters is a useful contribution. In addition, the authors also consider mis-specification of the scale of $\theta$. I would encourage the authors to be more precise in their comparison to prior work. This includes the experiments as well; in particular, Table 1 compares against an algorithm that assumes knowledge of $\tau$. But to my point above. How is this a fair comparison if assuming the scale of $\theta$ vs $\tau$ are essentially interchangeable? Presumably, Standard IGT only assumes $\tau$ and not the scale of $\theta$ as well.

**Requested Changes:**

Critical:
- Please clarify what you mean by "resolves the scale ambiguity" problem and discussion your contribution in the context of prior work that assumes the scale of $\tau$ rather than the scale of $\theta$. How are these fundamentally different?

Strengthen:
- Figures 1 & 3: Please briefly explain in the caption that these experiments are over a set of randomly generated games.
- Two-player, Zero-sum & QRE uniqueness: Please mention the scope of the paper is limited to two-player, zero-sum games early on. The first mention I see is Section 3. QRE uniqueness and tractability is mentioned several times before that despite the fact that uniqueness and tractability do not hold in n-player, general-sum.
- Algorithms 1 & 2: Please include these in the main body. I believe you have 2 extra pages you can use.
- Markov games: Please state somewhere that you assume players play stationary policies. There always exists a stationary QRE, but there exist equilibrium solutions to entropy regularized Markov games that are history dependent, so it would help to disambiguate things here.
- Markov games experiments: Did you also sample $\theta*$ from a standard normal like in the matrix games? Please add this detail.
- Experimental $\tau$'s: Can you give some rationale for why you chose $\tau*=2.0$ and $\tau*=1.5$?

Minor:
- Section 1.1: ratio $Q/\tau$ should probably be $\theta/\tau$.
- Section 2: IGT extends IRL should probably be IGT extends IO/IRL?
- Section 3.1: Add a note that equation (4) is linear in Q, the game parameters. It is unclear what linearity property you are referring to when you say "system of $m + n - 2$ linear constraints". It's definitely not linear in the strategies nor $Q$ and $\tau$ jointly.
- Assumption 1: It could be useful to explain that setting $\phi(a, b)$ to a one-hot representation allows one to encode an entire payoff matrix of $mn$ distinct parameters. Payoff matrices with shared parameters can be more concisely represented using $\phi$ though.
- Assumption 2: It is stated after the assumption that "Assumption 2 is necessary for the exact recovery of the scales of both $\theta*$ and $\tau*$". But Assumption 2 provides the scale of $\theta*$, so it does not make sense to say it is necessary for it's recovery. Is it really "recovered" if it is given?

---

### Review · Reviewer_YnBM · 2026-03-20

**Summary Of Contributions:**

This paper introduces the Normalized Least-Squares estimator to recover the reward and temperature parameters of a quantal response equilibrium of a zero-sum matrix or Markov game (under some assumptions).  The paper proves a number of theorem about the estimator, e.g., finite sample bounds on the parameters, and demonstrates the robustness of the estimator on randomly generated games.

**Audience:**

Yes

**Audience Explanation:**

The paper's contributions appear to be novel, and are about a topic that is of interest to the community.

**Claims And Evidence:**

Yes

**Claims Explanation:**

The mathematical claims in the paper appear to be accurate.  I have some concerns about how the contributions are described, though.  For example, the abstract makes the claim that Blind-IGT is the first framework to jointly recover both the reward and temperature parameters, but, as the paper argues, this is impossible in general.  In my opinion, it is disingenuous to point to assuming knowledge of the temperature parameter as being a limitation of other work that is then solved by instead assuming knowledge of the norm of the reward vector five pages later.

Related to the above, the paper claims “extensive simulations confirm the statistical eﬃciency of NLS”.  I do believe that the experiments are useful and informative, but I disagree that they are “extensive” or that they confirm the theory in a general sense.  Randomly generated games often do not exhibit worst-case behavior for algorithms.

Similarly, I do not believe the experiments demonstrate empirically that the algorithm is robust to unknown dynamics in Markov games.  This is due in part to the games being small and randomly generated, but it is also due to how the samples are gathered per-state, as opposed to using trajectories through the game.  i.e., the experimental setup guarantees uniform coverage of all states, which is impractical as the number of states grows.

**Requested Changes:**

In my opinion, there needs to be discussion about the various assumptions being made.  This includes nuance in the experimental setup.  In particular, why they are necessary and what scenarios they preclude.  e.g., in the statement of Theorem 2 in the paper it is not at all clear until actually reading the proof that N being sufficiently large means it could be actually be huge.

Similarly, I believe that the description of the contributions and how they generalize need to be toned down and re-worded.

---

### Review · Reviewer_5BYN · 2026-03-23

**Summary Of Contributions:**

This paper proposes a method for recovering parameters governing games (2-player zero-sum matrix, or markov games) under fairly strong (but argued to be necessary)  assumptions on the structure of these rewards. The chief contribution is that the authors do not assume the temperature parameter is known a-priori, unlike prior work.

At a high level, it is a slight modification a well-known trick of taking the difference in log probabilities (essentially exploiting the iia property of the QRE) to convert the problem into a system of linear equations. Generally, this yields roughly $\mathcal{O}(n+m)$ equations. Even here, there could already be more unknowns (m*n sized payoff matrix), so the authors assume further that each outcome has payoff assigned to $d$ known features, where $d$ is typically $<m+n-2$, and the payoffs is linear in these $d$ features --- this reduces the problem into learning $\theta$, the coefficient of this linear function.

What sets this paper apart is that temperature $\tau$ is not known, bilinearity arises from the temperature parameter, though this is much simpler than general bilinear problems since temperature is scalar.

The authors overcome the non-identifiability issue (scaling temperature and rewards can simply cancel themselves out) by assuming that  the unknown $\theta$ has a known norm $=1$. The second assumption is that the "data matrix" in the system of linear equations has full rank, such that if that the number of samples is large (and temperature $\tau$ known), the linear system would have a unique solution. The authors prove that under these assumptions (alongside other slightly weaker ones), with high probability that their method would recover the "true $\theta$". They also discuss what happens if their assumption on the norm of $\theta$ is not satisfied.

Finally, the authors show that their method applies to Markov Games as well.

**Additional Comments:**

N/A

**Audience:**

No

**Audience Explanation:**

I found the paper's novel contributions to be quite sparse even by TMLR's metrics. Fundamentally, the authors have essentially assumed away the non-identifiability problem by Assumptions 1, 2 and Theorem 1. Assumption 2 in particular essentially negates the impact of the unknown temperature parameter (one of the authors' main contributions) --- indeed, the algorithm in appendix 7 is precisely least squares  with a known temperature of 1 (but without the norm-of-$\theta$) constraint, but reporting the renormalized $\theta$ and scaling temperature to compensate.

Indeed, Figure 2 is in my opinion superfluous, is this not just a direct consequence of equation 10 (which is in turn directly from equation 7?), particularly if sampling error is not being addressed in the Figure?

As a matter of principle, it would seem that past equation 7, the rest of the paper has little to do with games, but rather an exercise in applying known results in perturbation theory. For instance, the assumption of a “soft-min” gap (Assumption 3) is standard requirement perturbation theory when studying systems $Ax=b$ with noisy $b$.

Even in the case of Markov games the algorithm is essentially converted into two steps, the first being to compute the Q function (exactly the same as before) before recovering the reward function as a separate step. In fact, the assumption that it is $Q^*$ (and not $r$) that is linear in $\theta$ is extremely strong and difficult to justify, but necessary for the logic in matrix games to follow through.

It also appears that more assumptions are made that are necessary. For instance, the theorems depend on the smallest singular value of $X^\*$, as far as I can tell, this is partly used to ensure (using Weyl’s inequality) that the sampled matrix $\hat{X}$ has full rank (section C3). However, could there not be a high probability statement that can be made about rank without resorting to this assumption? If $\hat{X}$ is a “noisy” version of $X^*$, I find it hard to believe that $\hat{X}$ is not full rank. Of  course, having a not-too-small singular value helps in bounding error (typically one would resort to the condition number of $\hat{X}$, but as far as rank goes, I don’t see an issue.

Finally, the authors have also missed out a fair amount of related work. For instance, the paper “Three Strategies to Success: Learning Adversary Models in Security Games” by Haghtalab et. al. solve a very similar system (albeit in a Stackelberg security setting) and analyze similar sample complexity bounds.

There is also some slightly misrepresentation of related work, particularly in inverse game theory. Work for example, by Waugh et. al. (2011) *are* computationally tractable, and non-uniqueness is not an issue if the assumption of maximum entropy is assumed — note that this paper essentially sidesteps the problem by assuming quantal responses (which for low temperatures is exactly the maximum entropy principle). Likewise, “complex bilevel optimization” is required precisely because the solution concept in those works are different, while the authors have selected the simplest model possible (2-player zero-sum games with entropy regularization/QRE).

In a similar vein, the authors should have made it clear from the beginning that they have made very strong assumptions (particularly in the norm of $\theta$). In particular, their characterization in Section 1.1 “...we cannot distinguish between a highly rational agent (small $\tau$) playing a low-stakes games (small $\theta$) and a highly irrational agent (large $\tau$) playing a high stakes games (large $\theta$)” makes it appear as if they have solved the multiplicative scale ambiguity, when in fact, by assuming that $||\theta||$ is known, they have essentially fixed the stakes of the game and assumed away the problem.

**Claims And Evidence:**

Yes

**Claims Explanation:**

Due to the sheer length of this paper (particularly the appendices), it was difficult to verify every single theorem by hand, however, I believe the overarching proof techniques *look sound on the surface*.

I also note that the experiments were run on extremely small environments (e.g., 10 $\times$ 10 matrix games, MGs with 8 states and 5 actions per state). While this is essentially a learning-theory paper (rather than game theory one) that deemphasizes experimental verification, "real" games, even ones in toy environments (e.g., Markov Soccer by Michael Littman) often exhibit precisely the pathologies that exist in the paper (e.g., states that are reached with near 0 probability under NE equilibria, actions that are dominated etc), that will result in constants in the paper to be too small/large (e.g., Assumption 3), so it would have been good to see what happens in some of these settings; it is typically understood that synthetic games can yield non-representative empirical results.

**Requested Changes:**

At the very minimum, the authors should not overstate their contribution in resolving the scale ambiguity (see above comments). Much more related work in inverse game theory should be referenced, in addition to the paper by Haghtalab et. al.

---

> ### Author Response · Authors · 2026-03-23
> **Rebuttal to Reviewer 5BYN**
>
> Dear Reviewer 5BYN,
>
> Thank you for the careful and technically engaged review. We appreciate that you identified the central structural issue correctly. Once the temperature parameter is unknown, the inverse-QRE problem is no longer the standard linear inverse problem studied in prior work. We will sharpen the wording so that the role of the normalization assumption is stated explicitly and early, but we would like to clarify several places where we believe the review collapses distinct issues: the algebraic reduction, the identifiability of the statistical model, and the actual target of inference.
>
> Your main criticism is that Assumption 2 **essentially negates** the unknown temperature parameter and reduces the problem to least squares with temperature fixed to $1$. We **respectfully disagree**. The fundamental issue is the positive scaling symmetry $(\theta^\ast,\tau^\ast) \mapsto (c\theta^\ast,c\tau^\ast)$, $c>0$, which leaves the induced QRE unchanged and is explicit in the inverse system
> $$X(\mu^\ast,\nu^\ast)\theta^\ast=\tau^\ast y(\mu^\ast,\nu^\ast).$$
> Without external calibration, exact point identification of $(\theta^\ast,\tau^\ast)$ is impossible. Assumption 2 does **not assume away** this fact — it is the scale calibration needed to break the gauge freedom. Prior QRE-based IGT fixes $\tau$ externally; Blind-IGT treats $\tau$ as a latent structural parameter that is part of the estimand.
>
> If one defines $z^\ast:=\theta^\ast/\tau^\ast$, Eq. (7) becomes $X^\ast z^\ast = y^\ast$, so the least-squares step estimates $z^\ast$, not $\theta^\ast$. Under Assumption 2, $\lVert\theta^\ast\rVert_2=C$, one obtains
> $$\tau^\ast=\frac{C}{\lVert z^\ast\rVert_2}, \qquad \theta^\ast=\frac{C}{\lVert z^\ast\rVert_2}z^\ast.$$
> Theorem 1 proves this lift is identifiable under rank and non-uniformity conditions; Theorem 2 proves the empirical lift is stable under sampling noise. The contribution is proving when calibrated joint recovery is possible and how error propagates through the nonlinear normalization map.
>
> This addresses the claim that the paper is **least squares with temperature fixed to $1$**. That is **not correct** as a statement about the inferential target. If one only cared about $z^\ast=\theta^\ast/\tau^\ast$, the problem reduces to a standard linear inverse problem. But Blind-IGT studies recovery of the calibrated pair $(\theta^\ast,\tau^\ast)$, which is exactly where the unknown temperature enters.
>
> Your comment that fixing $\lVert\theta\rVert$ is equivalent to fixing $\tau$ is **only partly correct**. Both break the scale ambiguity, but they differ as inferential formulations. Fixing $\tau$ removes bounded rationality from inference. Fixing $\lVert\theta\rVert_2=C$ leaves $\tau$ latent and estimable from behavior. They are equivalent as gauge choices, but **not as inverse problems or estimands**. This is why Figure 2 is not tautological: it asks whether misspecifying $\bar C$ destroys directional consistency when $\hat z$ is noisy. The answer is no, this is consistent with Lemma 3.
>
> We also **disagree** that past Eq. (7) the paper **has little to do with games**. The matrix $X(\mu^\ast,\nu^\ast)$ and vector $y(\mu^\ast,\nu^\ast)$ are induced by the coupled QRE fixed-point equations, not arbitrary regression objects. Recovering the structural parameter requires solving a system whose coefficients depend on equilibrium behavior. Reducing a structural problem to linear algebra after the correct transformation means the structure has been exploited, not that it has disappeared.
>
> On the Markov-game assumption: the inverse-softmax relation is a statement about $Q$-differences,
> $$\log \mu(a|s)-\log \mu(a'|s) = \frac{Q^\mu(s,a)-Q^\mu(s,a')}{\tau},$$
> so the natural linear object is $Q$, not $r$. This follows the same modeling logic as **Decoding Rewards (Liao et al.)**, where the inverse problem is formulated through linearly parameterized $Q$-objects which is the natural dynamic analogue of linear payoff parameterization in matrix games.
>
> Finally, the role of $\sigma_{\min}(X^\ast)$ is **not merely** to guarantee rank of $\(\hat X\)$. Our theorem is quantitative. Perturbation bounds scale as inverse powers of $\sigma_{\min}(X^\ast)$, so even a full-rank $\hat X$ yields an ill-conditioned inverse problem if $\sigma_{\min}(X^\ast)$ is tiny. Stability of inversion is what the lower bound controls.
>
> We will also expand related work, including the security-games paper you mention, and sharpen the distinction between Stackelberg security settings and simultaneous-play QRE-based inverse game theory.
>
> Thank you again for the thoughtful review.

---

### Author Response · Authors · 2026-04-01
**Delay In Rebuttals and Updating Paper**

Dear Action Editor and Reviewers,

First, we want to thank the reviewers for taking the time to provide such constructive and helpful feedback on our work.

We are leaving this quick note to give a heads-up about our timeline for the revisions. As the authors are currently students, we are right in the middle of our university exam period. Because of this, we've been having some trouble finding the focused time needed to properly implement all suggestions made by the reviewers and write out detailed replies to each of them.

We just wanted to keep everyone in the loop! We are actively working on the updates in the background and will get our formal responses and the revised PDF posted as soon as possible once our exams wrap up.

Thanks so much for your patience and understanding.

---

### Note · Authors · 2026-04-12

**Comment:**

Thank you to the reviewers and AE for the time and thoughtful feedback. We are withdrawing this submission solely because the authors are too busy with school work and cannot work on revisions properly right now without needing more time. We do not want to keep everyone waiting. We plan to revisit and resubmit later after addressing the suggested changes.

**Withdrawal Confirmation:**

I have read and agree with the venue's withdrawal policy on behalf of myself and my co-authors.